# CARP: Causal Alignment of Reward Models via Response-to-Prompt Prediction

## Abstract

Reward models (RMs) are central to aligning large language models (LLMs) with human preferences, yet they often overfit to spurious correlations such as response length or sycophancy. Existing approaches mainly focus on mitigating these artifacts, but overlook reinforcing the true causal link from prompt intentions to responses. We propose CARP (Causal Alignment of Reward Models via Response-to-Prompt Prediction), a framework that leverages inverse prompt prediction to measure how well a response addresses the intent embedded in its prompt. A prompt decoder is trained to estimate the original prompt embedding from a given response, and the reconstruction error defines a Semantic Alignment Score (SAS), which we use to adjust preference labels and regularize reward model training. We show theoretically that SAS isolates the prompt-to-response causal signal while filtering out spurious cues. Empirically, the prompt decoder selects shorter and less sycophantic responses with 87.7% accuracy across math, helpfulness, and safety benchmarks. Incorporating SAS into Bradley–Terry reward model training on Gemma-2B-it and Gemma-2-9B-it leads to significant improvements in RewardBench evaluation accuracy, demonstrating CARP's effectiveness in building more causally aligned reward models.

## 1 Introduction

Reinforcement Learning from Human Feedback (RLHF) has become a widely adopted framework for aligning large language models (LLMs) with human preferences (Christiano et al., 2023). A central component of this framework is the reward model, which is typically trained on pairwise human preference data to approximate evaluative judgments of model outputs and guide reinforcement learning towards outputs better aligned with human expectations(Ouyang et al., 2022).

However, recent work has revealed that reward models are susceptible to reward hacking, where models exploit imperfections in the learned reward function rather than genuinely aligning with human intent (Amodei et al., 2016). Reward hacking can arise from unintentional and prompt-irrelevant human preferences (Wang et al., 2025). For example, a preference for longer or sycophantic responses induces length bias(Stiennon et al., 2022) and sycophancy bias(Perez et al., 2022).

Early work focused on identifying specific spurious attributes and mitigating their impact on reward models. Shen et al. address length bias by decomposing the reward and suppressing the length-based bias signal during optimization. Later, causal methods (Pearl, 2009; Yao et al., 2021) were introduced to handle general unintentional artifacts. Some approaches reduce reward hacking by eliminating the causal edge from spurious artifacts to reward models; for instance, RRM attenuates this effect via counter-artifact data augmentation (Liu et al., 2025). In contrast, methods like CROME strengthen the causal edge from context-related intentions by generating augmented training samples (Srivastava et al., 2025). However, these methods only rely on data augmentation rather than explicitly quantifying prompt intentions in responses. We instead estimate this signal and use it to strengthen the causal edge from prompt intention to the reward model.

Estimating how much a given response faithfully reflects the prompt intention is difficult. The intention is a latent and unobservable variable. To capture such hidden factors, representation learning methods are often employed, such as sparse autoencoders (SAEs)(Makhzani & Frey, 2014) and variational autoencoders (VAEs)(Kingma & Welling, 2022). It also requires disentangling meaningful

alignment from incidental correlations and irrelevant attributes. Moreover, leveraging the prompt intention signal requires an effective mechanism to integrate it into reward model training.

To resolve these challenges, we frame reward model training within a causal graph to separate prompt-related intentions from context-free artifacts, develop a framework that quantifies how well a response realizes the latent prompt intention, and utilize it in training reward models. The pipeline is illustrated in Figure 1. To summarize, the contributions of this paper are three-fold:

- We point out that existing alignment studies lack frameworks to quantify a response's realization of prompt intention, particularly through causal manners.

- To address this, we construct a causal graph for reward model training, develop a framework (**CARP**) to quantify the extent of a response's alignment with prompt intention through **SAS**, and reinforce the causal effect of prompt intention in reward model training.

- We theoretically prove that SAS isolates prompt intention while compressing spurious artifacts. On RewardBench (Malik et al., 2025), our SAS-regularized reward model improves accuracy by 3.6% over the vanilla RM and RRM (Liu et al., 2025) on the 9B model.

- Downstream evaluations in Table 3 and Table 8 show that our model consistently favors on-topic responses, positioning CARP as a complementary component to nearly **all** existing reward hacking mitigation approaches and suggesting potential for further gains when integrated into a unified framework.

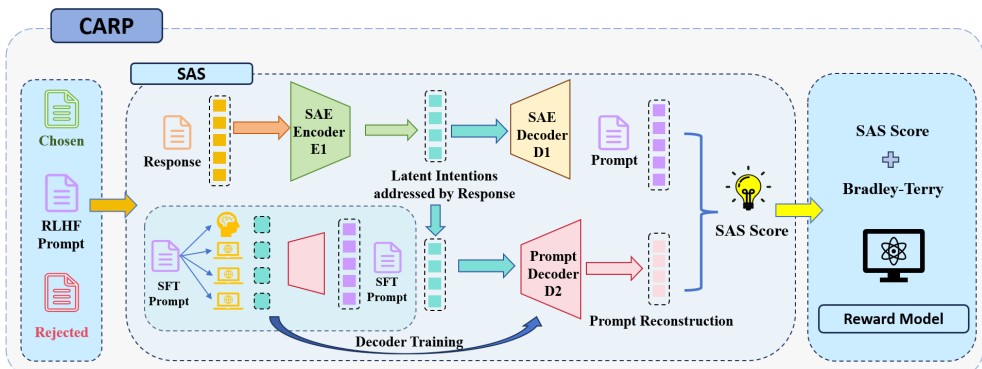

Figure 1: **CARP.** A prompt decoder is trained on multiple-response-to-one-prompt SFT data to suppress spurious signals. The resulting Semantic Alignment Score (SAS) is used as an additional signal in reward model training, incorporated into the loss function to strengthen the causal link between prompt intent and reward labels. This encourages the reward model to capture human preferences that are genuinely aligned with the prompt's intent.

## 2 SAS-REGULARIZED REWARD MODEL TRAINING

### 2.1 PROMPT-AWARE CAUSAL ABSTRACTION

Traditional methods typically build a causal graph as (Figure 2a), constructing $S$ and $C$ as effects of $X$ and $Y$, focusing on mitigating the causal effect from $C$ to $R$ ((Liu et al., 2025)). In contrast, we adopt an innovative modeling approach and formulate a DAG $\mathcal{G}$ to model the causal relationships (Figure 2b).

In $\mathcal{G}$, $X$ is the prompt, $Y$ is the response. $W \in \mathbb{R}^{d_w}$ is the latent human intention embedded within the prompt, which we assume to be the sufficient statistic that captures all human intentions from the prompt to generate the response. $Z \in \mathbb{R}^{d_z}$ is the latent artifact which we assume to be the sufficient statistic that captures all context-free causal factors that are necessary for generating a response, aside from $W$. We assume that $W$ is independent from $X$ and $W$. $R \in \mathbb{R}$ is the reward model.

Unlike traditional methods, our objective is to assign higher rewards to responses that are more aligned with the prompt's intention. Therefore, in our modeling, we employ anti-causal engineering

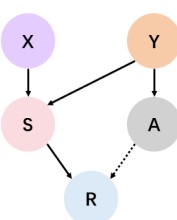

(a) Traditional causal graph of reward model. X is the prompt. Y is the response. S is the contextual signal that depends on X and Y. A is the context-free artifact that only depends on Y. R is the reward model.

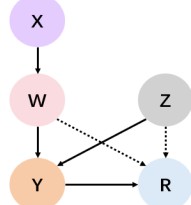

(b) CARP causal graph of reward model. X is the prompt. W is the human intention embedded within the prompt. Z is the context-free artifact that is independent from W and X. Y is the response. R is the reward model. Our work aims to build and strengthen the edge from W to R.

Figure 2: Causal graphs of Reward model.

to construct representations of latent $W$ and $Z$, while establishing and strengthening the causal edge from $W$ to $R$ via data augmentation. This encourages the reward model to preferentially capture responses aligned with the prompt's intention, thereby mitigating reward hacking.

**Setup** Suppose that we have a dataset of $N$ prompts with $M$ responses each. For the $i^{th}$ prompt and its $j^{th}$ response:

- **Prompt embedding**: $x_i \in \mathbb{R}^{d_x}$  • **Prompt intention**: $w_i = w(x_i) \in \mathbb{R}^{d_w}$
- **Artifacts**: $z_{i,j} \in \mathbb{R}^{d_z}$
- **Response embedding**: $y_{i,j} = f(w_i) + g(z_{i,j}) \in \mathbb{R}^d$ (Assume decomposed additivity)
- **Response SAE**: $\text{Encoder}(y_{i,j}) = u_{ij} = \text{TopK}(Py_{i,j})$, where $P \in \mathbb{R}^{k \times d}$
- **Prompt Decoder**: $\text{Decoder}(u_{i,j}) = Lu_{i,j} + b$, where $L \in \mathbb{R}^{d_x \times k}$

## 2.2 SEMANTIC ALIGNMENT SCORE (SAS)

Our key intuition is that *a decoder should be able to reconstruct the embedding of the prompt from the response representation if a response faithfully addresses the intent of its prompt.* Moreover, when multiple responses correspond to the same prompt, their shared components are more likely to capture the underlying intent, while spurious artifacts, such as verbosity or sycophancy, vary idiosyncratically and cancel out in expectation. We theoretically justified our ideas in Theorem 1 and Theorem 2. In practice, we train a prompt decoder that maps sparse response representations to their corresponding dense prompt embeddings. The training procedure consists of three stages: dataset preparation, representation extraction, and supervised decoder fitting.

**Data Construction** We build a hybrid 20K prompt–response pairs from two SFT corpora: Smoltalk (Allal et al., 2025) for reasoning and code tasks and AlpacaFarm (Dubois et al., 2024) for daily dialogues, and augment each prompt with three completions from DeepSeek-V3.1-Base (DeepSeek-AI et al., 2025), LLaMA3-72B (Grattafiori et al., 2024) and Qwen3-235B-A22B (Yang et al., 2025). Thus, each prompt has four responses, balancing semantic overlap and stylistic diversity to support learning invariant causal patterns.

**Representation Extraction** For each response, we extract a sparse semantic representation using the a sparse autoencoder (SAE) pretrained on LLaMA-3-8B[1] with TopK = 192 activation selection. This sparse vector serves as the input to the prompt decoder. The target output for the decoder is the last-token prompt embedding extracted from the 14th hidden layer of LLaMA-3-8B, which we treat as a stable and informative representation of the prompt's semantics.

---

[1] We used https://huggingface.co/EleutherAI/sae-llama-3-8b-32x

**Prompt Decoder Training**   Now the training proceeds by minimizing the Mean Squared Error (MSE) between predicted and target embeddings:

$$\mathcal{L}_{\text{pd}} = \arg\min_{L,b} \frac{1}{NM} \sum_{i=1}^{N} \sum_{j=1}^{M} \|Lu_{i,j} + b - x_i\|_2^2,$$

where $N$ is the number of prompts and $M$ is the number of responses per prompt. Given a response $u$ and a prompt $x$, We define the corresponding **Semantic Alignment Score (SAS)** as the reconstruction error, so a *lower SAS value indicates better alignment*:

$$\text{SAS}(u, x) = \|\hat{L}u + \hat{b} - x\|_2^2$$

## 2.3   THEORETICAL ANALYSIS OF SAS

We show that, with high probability, the output of our prompt decoder depends primarily on $w$ and $x$, and is approximately independent from $z$. This implies that **SAS evaluates how well a response aligns with the prompt's intent, compressing signals from artifacts.** For large $N$ and $M$, Theorem 1 states that the decoder parameters approximate the ideal ones that are independent from artifacts $z$. Meanwhile, Theorem 2 asserts that given a new sample response, the prompt decoder prediction is nearly independent from $z$. Theoretical support is provided below with formal proofs in Appendix B.

**Definition 1** (Ideal Top-K Indices). *The ideal case is that the decoder output **only contains** $w$ and is **independent from z**. For a given prompt intention $w_i$ and its corresponding signal $s_i = Pf(w_i) \in \mathbb{R}^k$, the **ideal Top-K indices** are defined as:*

$$J_{w_i} = \{j_1, j_2, \ldots, j_K\} \subset \{1, 2, \ldots, k\} \tag{1}$$

*where $j_1, j_2, \ldots, j_K$ are the indices corresponding to the K largest absolute values in $s_i = Pf(w_i)$. That is:*

$$|s_{i,j_1}| \geq |s_{i,j_2}| \geq \cdots \geq |s_{i,j_K}| \geq \max_{t \notin J_{w_i}} |s_{i,t}| \tag{2}$$

*Denote $I_{J_w}$ as the coordinate selection matrix corresponding to $J_w$, $I_{J_{real}}$ as the real coordinate selection matrix when choosing Top-K indices from $Py_{ij}$. Thus, we have:*

$$TopK(Py_{ij}) = I_{J_{real}} Py_{ij}, \quad TopK_{ideal}(Py_{ij}) = I_{J_w} Py_{ij}$$

**Definition 2** (Flip Event). *Given a prompt $i$ with ideal signal $s_i = Pf(w_i)$ and perturbation $\eta_{i,j} = g(z_{i,j})$, a **flip event** occurs when $TopK(P(f(w_i) + \eta_{i,j})) \neq J_{w_i}$.*

$$p_{\text{flip}} = \Pr(TopK(P(f(w_i) + \eta_{i,j})) \neq J_{w_i}) \tag{3}$$

**Definition 3** (Ideal Population Matrix). *The following matrices only depends on $w$ while independent form $z$.*

$$\Sigma_{xu}^{(0)} = \mathbb{E}[x(I_{J_w} s)^T], \quad \Sigma_{uu}^{(0)} = \mathbb{E}[(I_{J_w} s)(I_{J_w} s)^T]$$

$$L^{(0)} = \Sigma_{xu}^{(0)} (\Sigma_{uu}^{(0)})^{-1}, \quad b^{(0)} = \mathbb{E}[x] - L^{(0)} \mathbb{E}[I_{J_w} s]$$

**Theorem 1** (High-Probability Artifacts Suppression in Decoder). *Under assumptions (1)–(5) stated below, if $NM \geq C\frac{\sigma^2}{\varepsilon^2}(d + k + \log(1/\eta))$, then with probability at least $1 - \eta$, $\exists C_1, C_2 > 0$, such that:*

$$\|\widehat{L} - L^{(0)}\|_{op} \leq C_1(\varepsilon + p_{\text{flip}}), \quad \|\hat{b} - b^{(0)}\|_2 \leq C_2(\varepsilon + p_{\text{flip}})$$

**Theorem 2** (Artifacts Suppression in Prediction). *Under Assumptions (1)–(5) stated in Appendix B, given a new sample $y = f(w) + g(z)$, $u_{new} = TopK(Py)$, then for any confidence parameter $\eta \in (0, 1)$, with probability at least $1 - \eta$ the following holds:*

$$\left\|\widehat{L}u_{\text{new}} + \widehat{b} - (L^{(0)} I_{J_w} Pf(w) + b^{(0)})\right\|_2$$
$$\leq \widetilde{C}\left((\varepsilon + p_{\text{flip}}) \|P\|_{\text{op}} \frac{M_f}{\sqrt{\eta}} + \sigma\sqrt{k + \log(1/\eta)}\right), \tag{4}$$

*where $\sigma$ is the sub-Gaussian scale according to assumption 2 in Appendix B, and $\widetilde{C} > 0$ is a constant depending only on the constants appearing in Assumptions (1)–(5) and on operator norms of $L^{(0)}$ and $P_{J_w}$.*

## 2.4 SAS-REGULARIZED DYNAMICS IN REWARD MODEL TRAINING

We extend the Bradley–Terry framework with SAS regularization. Let $r_c, r_r$ be the reward scores of the chosen and rejected responses, $s_c, s_r$ their SAS scores, $\sigma$ the sigmoid function and $k$ the tuning parameter. The loss of vanilla RLHF and SAS-based RLHF are as follows:

$$\mathcal{L}_{\text{vanila}} = -\sum_i \log \sigma(y_{ic} - y_{ir}), \quad \mathcal{L}_{\text{SAS}} = -\sum_i \log \sigma\big((y_{ic} - y_{ir}) + k \cdot (s_{ic} - s_{ir})\big) \quad (5)$$

$$\hat{r}_n(x, y) = \arg\max_r[-L_{vanilla}], \quad \hat{r}_{nSAS}(x, y) = \arg\max_r\big[-L_{SAS}\big] \quad (6)$$

**Effect on Parameter Updates**   Here we analyze the effect of SAS-regularized training process through gradients in the parameter updates, with detailed derivations provided in Appendix C.1.

Since we have

$$\frac{\partial L}{\partial \theta} = \sum_i [\sigma(y_{ic} - y_{ir}) - 1]\left[\frac{\partial y_{ic}}{\partial \theta} - \frac{\partial y_{ir}}{\partial \theta}\right]$$

$$\frac{\partial L_{SAS}}{\partial \theta} = \sum_i [\sigma(y_{ic} - y_{ir} + k(s_{ic} - s_{ir})) - 1]\left[\frac{\partial y_{ic}}{\partial \theta} - \frac{\partial y_{ir}}{\partial \theta}\right]$$

SAS modulates gradients: when aligned with preferences, it magnifies updates toward prompt intention; when in conflict, it mitigates them, thus modifying the update steps and reducing artifact influence.

**Causal Nature of SAS**   According to Proposition 1 in Appendix C.2, we have $\hat{r}_{nSAS}(x, y) = \hat{r}_n(x, y) - k \cdot s(x, y)$.

We evaluate the causal effect of SAS by deriving the ATE on the difference between on-intention and off-intention responses, where the treatment corresponds to incorporating SAS rather than the presence of intention itself.:

$$ATE = \mathbb{E}[\hat{r}(x, y_{on}) - \hat{r}(x, y_{off})|SAS] - \mathbb{E}[\hat{r}(x, y_{on}) - \hat{r}(x, y_{off})|vanilla]$$
$$= \mathbb{E}[\hat{r}_{nSAS}(x, y_{on}) - \hat{r}_{nSAS}(x, y_{off})] - \mathbb{E}[\hat{r}_n(x, y_{on}) - \hat{r}_n(x, y_{off})]$$
$$= k\mathbb{E}[-s(x, y_{on}) + s(x, y_{off})] \geq 0$$

Therefore, although SAS can be regarded as a penalty term for the reward, it induces a positive shift in the reward difference between on-intention and off-intention responses compared to vanilla. Consequently, incorporating SAS effectively strengthens the causal effect of prompt intention signal on the reward model. In Section 2.3, we show that in high probability, the decoder output is approximately independent from artifact $z$, so do SAS. **Thus, the causal effect introduced by SAS is independent from $z$, thereby removing $z$ as a confounder.**

**Curriculum Learning Schedule**   To facilitate stable training, we implement a curriculum learning approach so that $k_{\text{eff}} = k \cdot I(\text{Epoch} \geq 1)$.

**Safety Alignment Considerations**   Denote $\delta_{\text{sas}} = s_c - s_r$. In practice, we apply thresholding for safety alignment scenarios. Safety-critical cases often exhibit counterintuitive SAS patterns where safe responses (e.g., refusal to answer harmful queries) may appear "off-topic" compared to potentially dangerous but directly responsive answers. To handle this, we introduce a safety threshold $\tau$: $\delta_{\text{sas}}^{\text{thres}} = \delta_{\text{sas}}\mathbf{1}(\delta_{\text{sas}} \leq \tau)$. When $\delta_{\text{sas}} > \tau$, the SAS regularization is disabled ($\delta_{\text{sas}}^{\text{thres}} = 0$), allowing the loss to revert to standard Bradley–Terry preference learning. This mechanism preserves safety alignment by preventing SAS scores from interfering with cases where topical deviation may actually indicate safer, more appropriate responses. We demonstrated the effectiveness of our thresholding further in Section 3.2.

# 3 EXPERIMENTS

In this section, we first systematically evaluate the overall performance of the prompt decoder trained using the scheme described in Section 2.2. We then visualize the distribution of the computed SAS scores on the RLHF training set, and finally present the downstream reward model training results.

## 3.1 PROMPT DECODER RESULTS

**Evaluation Dataset.** We construct a 300-sample evaluation set by sampling 100 preference pairs from each of the following sources: (i) 100 pairs of helpfulness preference from the HH-RLHF-Helpful-standard (Dong et al., 2024). (ii) 100 pairs from the Reward-Bench-2 (Malik et al., 2025) math category. (iii) 100 pairs from the Reward-Bench-2 safety category. To evaluate the sensitivity of the prompt decoder to stylistic artifacts, we create perturbed versions of the chosen responses using the GPT-4o-mini model. The rewriting prompt is designed to preserve the factual content while introducing stylistic variations; detailed rewrite instructions are demonstrated in the Appendix D.

To validate Theorem 1, we first train a decoder on a dataset of 20K prompts *without augmentation*. As shown in Figure 3a, the decoder already achieves solid performance: selects the human preferred response over its stylistic rewrite in roughly 80% cases, where selection means having a lower SAS score, indicating that the decoder has successfully learned to filter out superficial stylistic variations.

To further verify the effectiveness of the one-to-many training paradigm, we compare three settings: (i) 20K without augmentation, (ii) 20K with four responses per prompt (augmented), and (iii) 80K unaugmented prompts, which matches the augmented setting in total number of responses. We evaluate each decoder along two axes: (1) distinguishing chosen from rewritten responses, and (2) distinguishing chosen from rejected responses. The results are presented in Figure 3a and Figure 3b.

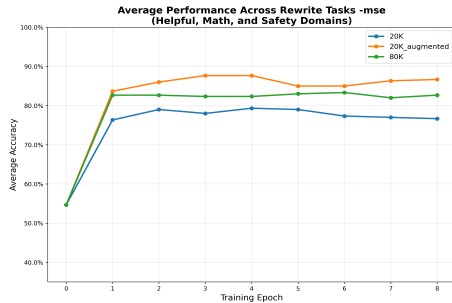

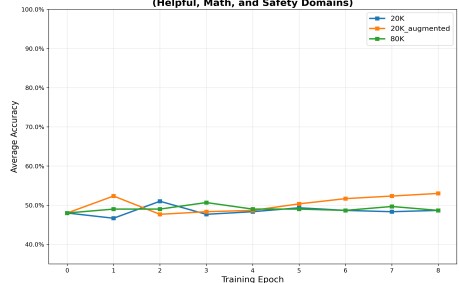

(a) Average accuracy of the prompt decoder on the chosen-vs-rewrite task across helpful, math, and safety domains. Augmented training (20K_augmented) yields the best performance, surpassing both unaugmented 20K and 80K data.

(b) Average accuracy of the prompt decoder on the chosen-vs-reject task. Performance remains near random guess (50%) across all training regimes, indicating that SAS captures a signal orthogonal to human preference labels.

Figure 3: Average Accuracy Curve of Prompt Decoder

All prompt decoders were trained with a batch size of 128 and a learning rate of $1e-5$ for 8 epochs on a single NVIDIA RTX 4090 GPU. Each decoder matches the size of the encoder used in the corresponding sparse autoencoder (SAE) mentioned in 2.2. Across all epochs, the augmented 20K dataset achieves highest accuracy $87.7\%$ and outperforms both the 20K and 80K baselines on the chosen-vs-rewrite task, indicating that response augmentation offers stronger supervision than simply increasing data volume. In particular, the decoder consistently fails to distinguish chosen from rejected responses, with accuracy near 50% regardless of the size of the data set. This highlights that SAS is a complementary alignment signal rather than leaking human preference supervision, and thus further *filtering out unintentional signal introduced by human labellers*.

## 3.2 REWARD MODEL RESULTS

**Training and Evaluation Datasets.** We follow the training and evaluation protocol established in RRM (Liu et al., 2025). For training, we randomly sample a 70K subset from their 700K

| Prompt Decoder | Chosen vs Rewrite ($\uparrow$) | | | | Chosen vs Reject ($\rightarrow$50%) | | | |
|---|---|---|---|---|---|---|---|---|
| | Helpful | Math | Safety | Overall | Helpful | Math | Safety | Overall |
| 20K | 73.0 | 94.0 | 71.0 | 79.3 | 53.0 | 51.0 | 41.0 | 48.3 |
| 80K | 75.0 | **98.0** | 77.0 | 83.3 | 56.0 | 47.0 | 43.0 | 48.7 |
| 20K_augmented | **86.0** | 93.0 | **84.0** | **87.7** | 53.0 | 47.0 | 46.0 | 48.7 |

Table 1: Accuracy (%) of prompt decoders on the **Chosen vs Rewrite** and **Chosen vs Reject** tasks, evaluated at the best epoch for each model across helpful, math, and safety domains.

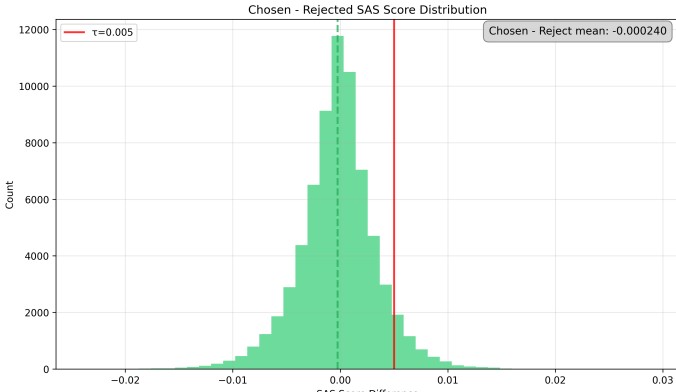

Figure 4: Distribution of the difference of Semantic Alignment Scores (SAS) between chosen and rejected responses on the 70K training pairs.

RLHF dataset (Dong et al., 2024)[2], which contains approximately 700K pairwise preference examples. While RRM uses a pairwise preference reward model (Jiang et al., 2023), we instead adopt a Bradley–Terry scheme (Bradley & Terry, 1952). For evaluation, we adopt RewardBench (Lambert et al., 2024), which provides curated test sets across four evaluation dimensions—*chat*, *chat-hard*, *safety*, and *reasoning*.

**SAS for Reward Model.** To compute SAS for the 70K training pairs, we use the prompt decoder trained on the 20K augmented dataset at Epoch 3, which achieves the highest accuracy on the chosen-vs-rewrite task (Table 1) and best reconstructs prompt embeddings from responses.

Once selected, the decoder remains frozen throughout reward model training. For each training pair $(x, y^+, y^-)$, we compute SAS scores by encoding the chosen and rejected responses into sparse vectors via the SAE, and decoding them back into the prompt embedding space. We visualize the distribution of SAS scores across the training set in Figure 4. While the chosen responses tend to have slightly lower SAS values than the rejected ones, the overall distributions are closely aligned. This observation motivates the use of a larger tuning parameter $k$ in the SAS-regularized loss (Equation 5) to amplify the effect of this fine-grained alignment signal during training.

**RM Training.** We fine-tune reward models based on Gemma-2-2B-it (Team et al., 2024) and Gemma-2-9B-it, using the SAS-regularized Bradley–Terry objective. Each model is trained for 2 epochs with a batch size of 256 and a learning rate of $2\mathrm{e}{-6}$, optimized using AdamW with cosine learning rate decay. We set $k = 0$ during the first epoch to allow the model to learn basic preference alignment, and apply non-zero SAS regularization only in the second epoch. All training is conducted on an $8\times$NVIDIA H200 GPU cluster. We experiment with $k \in \{4\mathrm{e}3,\ 1.6\mathrm{e}4,\ 3.2\mathrm{e}4,\ 6.4\mathrm{e}4\}$ and find that the best performance is achieved at $k = 3.2\mathrm{e}4$ for the 2B model and $k = 6.4\mathrm{e}4$ for the 9B model. For all subsequent reward model training, we set the safety threshold $\tau = 0.005$, which filters out approximately 7% of extreme cases from the training data, and results are shown in

[2]https://huggingface.co/datasets/RLHFlow/pair_preference_model_dataset

Table 2. On RewardBench, the overall accuracy of the 9B model improves from 83.22% to **86.83%**. For both 2B and 9B models, the *Chat-Hard* category sees a consistent gain of over **4%**. Detailed evaluations for each single scaling value are attached in Appendix D. We obtain the baseline model simply by setting $k = 0$. Moreover, we apply RRM's data permutation framework to the Bradley–Terry reward model. Details of the training setup are provided in the Appendix D.

**(a) Gemma-2B-it** ($k = 3.2e4$)

| Model | Chat | Chat-Hard | Safety | Reasoning | Avg. | Weighted Avg. |
|---|---|---|---|---|---|---|
| Vanilla RM | **97.77** | 54.82 | **83.24** | 66.18 | 75.50 | 72.46 |
| Bradley–Terry RRM | 92.19 | 48.03 | 49.46 | 69.11 | 64.69 | 63.79 |
| CARP (Ours) | 96.93 | **58.99** | 79.05 | **71.56** | **76.63** | **74.54** |

**(b) Gemma-9B-it** ($k = 6.4e4$)

| Model | Chat | Chat-Hard | Safety | Reasoning | Avg. | Weighted Avg. |
|---|---|---|---|---|---|---|
| Vanilla RM | 96.37 | 63.37 | **89.73** | 82.88 | 83.09 | 83.22 |
| Bradley–Terry RRM | 93.02 | 59.65 | 61.22 | 78.55 | 73.11 | 73.10 |
| CARP (Ours) | 94.69 | **68.86** | 88.24 | **89.87** | **85.42** | **86.83** |

Table 2: **RewardBench accuracy (%) of reward models across four evaluation categories.** CARP (Ours) denotes the SAS-regularized reward model with best-performing $k$ value. Each subtable corresponds to a different model scale. The weighted average reflects the overall proportion of correctly ranked preference pairs across all subsets.

**Safety Alignment**  We conduct an ablation study to assess the impact of the safety threshold $\tau$ when $k = 3.2e4$. As shown in Table 3, the model with thresholding ($\tau = 0.005$) outperforms the one without thresholding ($\tau = 0$) on the *Safety* dimension.

**Spurious Correlation Analysis.**  To further assess the robustness of our SAS-regularized reward models to spurious alignment signals, we conduct a subtle experiment on the same 300 preference pairs subsets sampled from RewardBench2 when we evaluate the prompt decoder 3.1. For each chosen response, we construct three rewrites designed to isolate specific confounding factors:

- **Rewrite 1 (Lengthened)**: We apply a `RATE`-style rewriting prompt to make the chosen response significantly longer, while preserving its factual content, stance, and topicality (Reber et al., 2025).

- **Rewrite 2 (Shortened)**: Starting from Rewrite 1, we apply another `RATE`-style prompt to reduce its length, again without altering the original intent or content.

- **Rewrite 3 (Lengthened, Off-topic)**: We generate a longer version of the chosen response that includes slight topical drift—maintaining politeness and fluency, but deviating from the core question or user intent.

By comparing the reward scores assigned to `Rewrite1 vs Rewrite2`, we test whether the reward model exhibits *length bias*—i.e., whether longer responses are consistently favored despite content parity. Meanwhile, comparing `Rewrite1 vs Rewrite3` probes the model's ability to penalize off-topic responses, even when they are longer or more stylistically polished.

This design ensures that any performance difference arises from the model's sensitivity to spurious features such as verbosity or topic coherence. Our results in Table 4 show that SAS-regularized models remains indifferent to length bias while being more sensitive to topical alignment.

We observe similar trends in the 9B setting 7, where CARP amplifies the distinction between on-topic and off-topic responses while remaining robust to verbosity.

| Model | Chat | Chat-Hard | Safety | Reasoning | Avg. | Weighted Avg. |
|---|---|---|---|---|---|---|
| CARP ($\tau = 0.005$) | 96.93 | 58.99 | **79.05** | 71.56 | 76.63 | 74.54 |
| CARP ($\tau = 0$) | 96.09 | 62.06 | 77.97 | 70.09 | 76.55 | 73.94 |

Table 3: RewardBench accuracy (%) comparison of best CARP 2B-model with and without SAS thresholding. Using thresholding ($k = 3.2e4, \tau = 0.005$) disables SAS regularization for safety-critical examples. We observe that removing the threshold ($\tau = 0$) reduces the model safety.

| Model (2B) | Rewrite1 vs Rewrite2 | | | | Rewrite1 vs Rewrite3($\uparrow$) | | | |
|---|---|---|---|---|---|---|---|---|
| | Helpful | Math | Safety | Avg. | Helpful | Math | Safety | Avg. |
| Vanilla RM | 43.0 | 55.0 | 59.0 | 52.33 | 57.0 | 92.0 | **90.0** | 79.67 |
| Bradley–Terry RRM | 44.0 | **74.0** | **62.0** | **60.0** | 53.0 | 62.0 | 86.0 | 72.0 |
| CARP(Ours) | **53.0** | 67.0 | 46.0 | 55.33 | **83.0** | **95.0** | 89.0 | **89.0** |

Table 4: Accuracy (%) of reward models on the **Rewrite1 vs Rewrite2** and **Rewrite1 vs Rewrite3** tasks, evaluated at the best epoch for each model across helpful, math, and safety domains.

## 4 CONCLUSION AND FUTURE DISCUSSION

Reward hacking arises from unintentional, prompt-unrelated biases in preference data. Prior work has sought to address this issue by reinforcing the causal link between prompt intent and reward model predictions, but has lacked a principled framework to quantify the extent to which a response aligns with the prompt. We propose **CARP**, a framework that introduces the **Semantic Alignment Score (SAS)** to measure how well a response reflects latent prompt intentions. We theoretically show that SAS depends only on prompt-relevant information and suppresses context-independent artifacts with high probability. Experimental results 5 and 5 show that SAS captures prompt intent independently of human preference labels. Incorporating SAS into reward model training further improves performance over both Vanilla RM and RRM. Our framework thus enables reward models to be more directly guided by prompt semantics, reducing reliance on spurious artifacts and mitigating reward hacking. Results in in Table 3 and Table 8 show that that CARP improves reward model behavior in a subtle, orthogonal manner to most existing reward hacking mitigation methods. Rather than replacing existing methods, CARP offers a principled mechanism for injecting prompt intent supervision into reward training, opening the door to unified pipelines.

## 5 RELATED WORK

**Reward Hacking**   The problem of reward hacking has become increasingly prominent with the growing adoption of RLHF (Amodei et al., 2016; Casper et al., 2023; Kaufmann et al., 2023). Models are likely to achieve high rewards without fulfilling the intended objectives(Pan et al., 2022; Weng, 2024). For example, reward models are easily hacked by lengthSinghal et al. (2024), sycophancyPerez et al. (2022), concept(Zhou et al., 2024), and demography(Salinas et al., 2023). Recent works employ model merging (WARP(Ramé et al., 2024a) and WARM(Ramé et al., 2024b)), and hacking reward decomposition(Chen et al., 2024) to mitigate hacking in online RLHF.

**Causal Solutions to Reward Hacking**   On one hand, some researchers weaken the causal edge from spurious attributes. RATE employs a "rewrite-twice" strategy to correct the imperfections of counterfactuals(Reber et al., 2025). RRM trains robust reward models by augmenting the training distribution with counter-artifact examples(Liu et al., 2025). Causal-Debias explicitly represents spurious attributes and trains invariant predictors by minimizing the dependence between learned representations and such attributes(Zhou et al., 2023). On the other hand, others enhance the causal relationship among intentional causal attributes. CROME applies causal data augmentation by intervening on causally relevant attributes to generate training samples, strengthening their influence on the reward model(Srivastava et al., 2025).

## REPRODUCIBILITY STATEMENT

All code used for training the response SAE, prompt decoder, and reward models (RM, RRM, and SAS-regularized RM), as well as for running the experiments, will be made publicly available upon publication. The full implementations of the data generation pipeline and the training procedures for SAE, prompt decoder, and reward models will be released on GitHub, and all trained models will be uploaded to HuggingFace. For publicly available datasets used in our experiments, we provide detailed preprocessing steps in the supplementary materials. For datasets generated by us, we will release them on HuggingFace, with rewriting prompts described in the appendix. We also include the complete set of hyperparameters (e.g., learning rates, batch sizes, and optimization settings) to facilitate replication. Finally, our evaluation protocols are fully documented in the main text and appendix, ensuring that all reported results can be reproduced.

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

## A  LLM USAGE STATEMENT

In preparing this manuscript, large language models (LLMs) were used solely as auxiliary tools for improving the clarity and readability of the text. Specifically, LLMs were employed to correct grammatical errors, refine phrasing, and polish the language style to ensure that the writing is more formal and consistent with academic standards.

Importantly, LLMs were not used for research ideation, retrieval or discovery of related work, data analysis, or generation of scientific content. All conceptual contributions, methodological designs, experimental implementations, and substantive writing were conducted entirely by the authors. The authors take full responsibility for the final content of the paper.

## B  THEORETICAL DERIVATION FOR ARTIFACTS COMPRESSION

**Assumption 1** (Conditional Zero-Mean of Artifacts)**.**
$$\mathbb{E}[g(z_{i,j}) \mid w_i] = 0.$$

*Since $z$ is independent from $w$, the conditional expectation is a constant, which can be generalized to non-zero case easily.*

**Assumption 2** (Sub-Gaussian Distribution)**.**

1. *There exist a constant $\sigma > 0$ such that for every coordinate of $p_r$ and every $\lambda \in \mathbb{R}$,*
$$\mathbb{E}\big[\exp(\lambda \cdot p_r^T g(z_{i,j}))\big] \leq \exp(\sigma^2 \lambda^2 / 2).$$

2. *There exist constants $\sigma_x, \sigma_y > 0$ such that for every unit vectors $a \in \mathbb{R}^{d_x}$, $b \in \mathbb{R}^{d_y}$, and every $\lambda \in \mathbb{R}$,*
$$\mathbb{E}\Big[\exp\Big(\lambda\, a^\top (x_i - \mu_x)\Big)\Big] \leq \exp\Big(\frac{\lambda^2 \sigma_x^2}{2}\Big),$$
$$\mathbb{E}\Big[\exp\Big(\lambda\, b^\top (u_{ij} - \mu_u(x_i))\Big)\Big] \leq \exp\Big(\frac{\lambda^2 \sigma_u^2}{2}\Big), \ \mu_u(x_i) := \mathbb{E}[u_{i,j} \mid x_i].$$

**Assumption 3** (Top-K Margin Condition)**.** *For $s_i = P f(w_i)$ and ideal Top-K indices $J_{w_i}$, there exists $\delta > 0$ such that:*
$$\min_{j \in J_{w_i}} \min_{t \notin J_{w_i}} (|s_{i,j}| - |s_{i,t}|) \geq \delta.$$

**Assumption 4** (Positive Definite Covariance)**.**
$$\lambda_{\min}(\Sigma_{uu}) \geq \lambda_0 > 0,$$
*where $\Sigma_{uu} = \mathbb{E}[uu^T]$.*

**Assumption 5** (Bounded Expectation)**.**
$$\exists M_x, M_f, M_u > 0, \ s.t. \ \mathbb{E}[\|x_i\|_2^2] \leq M_x^2, \ \mathbb{E}[\|f(w_i)\|_2^2] \leq M_f^2, \ \mathbb{E}[\|u_{ij}\|_2^2] \leq M_u^2$$

**Lemma 1.** *If $\|P\eta_{i,j}\|_\infty < \delta/2$, then no flip occurs. Moreover, the flip probability satisfies:*
$$p_{\text{flip}} \leq \Pr(\|P\eta_{i,j}\|_\infty \geq \delta/2) \leq 2k \exp\left(-\frac{\delta^2}{8\sigma^2}\right)$$

***Proof of Lemma 1.*** Let $s_i = P f(w_i)$ and $\Delta_{i,j} = P\eta_{i,j}$. For any $j \in J_{w_i}$ and $t \notin J_{w_i}$:
$$
\begin{aligned}
|(s_i + \Delta_{i,j})_j| - |(s_i + \Delta_{i,j})_t| &\geq |s_{i,j}| - |\Delta_{i,j,j}| - |s_{i,t}| - |\Delta_{i,j,t}| \\
&\geq |s_{i,j}| - |s_{i,t}| - 2\|\Delta_{i,j}\|_\infty \\
&\geq \delta - 2(\delta/2) = 0
\end{aligned}
$$

Thus the Top-K selection remains unchanged.

By union bound:

$$\Pr(\|P\eta_{i,j}\|_\infty \geq \delta/2) = \Pr\left(\max_{r=1,\dots,k} |p_r^T \eta_{i,j}| \geq \delta/2\right) \leq \sum_{r=1}^{k} \Pr(|p_r^T \eta_{i,j}| \geq \delta/2) \tag{7}$$

Since $p_r^T \eta_{i,j}$ is sub-Gaussian with parameter $\sigma^2$, by tail bounds:

$$\Pr(|p_r^T \eta_{i,j}| \geq \delta/2) \leq 2 \exp\left(-\frac{(\delta/2)^2}{2\sigma^2}\right) = 2\exp\left(-\frac{\delta^2}{8\sigma^2}\right) \tag{8}$$

Therefore: $p_{\text{flip}} \leq 2k\exp(-\delta^2/8\sigma^2)$. $\qquad\qquad\qquad\qquad\qquad\qquad\qquad\qquad\square$

**Lemma 2** (Population Covariance Decomposition). *Denote $I_{J_w}$ as the coordinate selection matrix corresponding to $J_w$, and $I_{\text{flip}}$ as the real coordinate selection matrix when flipping.*

$$\Sigma_{xu}^{(0)} = \mathbb{E}\big[x(I_{J_w}s)^T\big], \quad \Sigma_{xu} = \mathbb{E}\big[xu^T\big], \quad \Sigma_{uu}^{(0)} = \mathbb{E}\big[(I_{J_w}s)(I_{J_w}s)^T\big], \quad \Sigma_{uu} = \mathbb{E}\big[uu^T\big]$$

*The population cross-covariance can be decomposed as:*

$$\Sigma_{xu} = \Sigma_{xu}^{(0)} + \Delta_{xu}, \quad \Sigma_{uu} = \Sigma_{uu}^{(0)} + \Delta_{uu}$$

*with $\|\Delta_{xu}\|_{op} \leq C_x p_{\text{flip}}, \ \|\Delta_{uu}\|_{op} \leq C_u p_{\text{flip}}$.*

***Proof of Lemma 2.*** Recall the notation for a fixed prompt index $i$:

$$y_{i,j} = f(w_i) + g(z_{i,j}), \qquad v_{i,j} = Py_{i,j}, \qquad u_{i,j} = \text{TopK}(v_{i,j}),$$

and write, for brevity,

$$s_i := Pf(w_i), \qquad \eta_{i,j} := g(z_{i,j}), \qquad \Delta_{i,j} := P\eta_{i,j},$$

so that $v_{i,j} = s_i + \Delta_{i,j}$ and $u_{i,j} = \text{TopK}(s_i + \Delta_{i,j})$.

**Step 1**  Prove that there exist constants $C > 0$, such that $\|\mathbb{E}\big[\Delta_{ij} \mid w_i, \text{flip}\big]\|_2^2 \leq C$. Fix a coordinate $r \in \{1, \ldots, k\}$. Denote $\Delta_{ijr} := (P\eta_{i,j})_r = p_r^\top \eta_{i,j}$. By the sub-Gaussian assumption,

$$\mathbb{P}(|\Delta_{ijr}| \geq t) \leq 2\exp\left(-\frac{t^2}{2\sigma^2}\right), \quad \forall t > 0.$$

The flip event implies that the Top-K selection has been altered, which by the margin assumption requires

$$\text{flip} \implies |\Delta_{ijr}| \geq \delta/2 \quad \text{for some } r.$$

Hence, for each coordinate,

$$\mathbb{E}[\Delta_{ijr}^2 \mid w_i, \text{flip}] \leq \mathbb{E}[\Delta_{ijr}^2 \mid |\Delta_{ijr}| \geq \delta/2].$$

By the definition of conditional expectation,

$$\mathbb{E}[\Delta_{ijr}^2 \mid |\Delta_{ijr}| \geq \delta/2] = \frac{\mathbb{E}[\Delta_{ijr}^2 \mathbf{1}_{|\Delta_{ijr}| \geq \delta/2}]}{\mathbb{P}(|\Delta_{ijr}| \geq \delta/2)}.$$

Using the sub-Gaussian tail bound, the numerator can be bounded by integrating the tail:

$$\mathbb{E}[\Delta_{ijr}^2 \mathbf{1}_{|\Delta_{ijr}| \geq \frac{\delta}{2}}] = \int_0^\infty \mathbb{P}(\Delta_{ijr}^2 \mathbf{1}_{|\Delta_{ijr}| \geq \frac{\delta}{2}} \geq t)\, dt = \int_0^{\frac{\delta^2}{4}} 1\, dt + \int_{\frac{\delta^2}{4}}^\infty 2\exp\left(-\frac{t}{2\sigma^2}\right) dt \leq C_1\sigma^2,$$

where $C_1 > 0$ is a constant depending only on $\delta$ and $\sigma$. The denominator $\mathbb{P}(|\Delta_{ijr}| \geq \delta/2) \leq 1$, so

$$\mathbb{E}[\Delta_{ijr}^2 \mid |\Delta_{ijr}| \geq \delta/2] \leq C_1\sigma^2.$$

Finally, summing over all $k$ coordinates,

$$\|\mathbb{E}\big[\Delta_{ij} \mid w_i, \text{flip}\big]\|_2^2 \leq \mathbb{E}[\|\Delta_{i,j}\|_2^2 \mid w_i, \text{flip}] = \sum_{r=1}^k \mathbb{E}[\Delta_{ijr}^2 \mid w_i, \text{flip}] \leq kC_1\sigma^2,$$

which proves the conclusion with $C := kC_1\sigma^2$.

**Step 2** Prove the lemma.

Using the law of total expectation:

$$\mathbb{E}[\Delta_{ij} \mid w_i] = \mathbb{E}[\Delta_{ij} \mid w_i, \text{no-flip}]P(\text{no-flip} \mid w_i) + \mathbb{E}[\Delta_{ij} \mid w_i, \text{flip}]P(\text{flip} \mid w_i)$$
$$= \mathbb{E}[\Delta_{ij} \mid w_i, \text{no-flip}](1 - P(\text{flip} \mid w_i)) + \mathbb{E}[\Delta_{ij} \mid w_i, \text{flip}]P(\text{flip} \mid w_i)$$
$$= 0$$

$$\mathbb{E}[\Delta_{ij} \mid w_i, \text{no-flip}] = -\frac{P(\text{flip} \mid w_i)}{1 - P(\text{flip} \mid w_i)}\mathbb{E}[\Delta_{ij} \mid w_i, \text{flip}]$$

Here we denote $\tilde{\Delta} = \mathbb{E}[\Delta_{ij} \mid w_i, \text{flip}]$, according to **Step 1**, $||\tilde{\Delta}||_2^2 \le C$,

$$\mathbb{E}\big[u_{ij} \mid w_i\big]$$
$$= \mathbb{E}\big[u_{ij} \mid w_i, \text{no-flip}\big] \cdot P(\text{no-flip}|w_i) + \mathbb{E}\big[u_{ij} \mid w_i, \text{flip}\big] \cdot P(\text{flip}|w_i)$$
$$= (1 - P(\text{flip}|w_i))I_{J_{w_i}}(s_i + \mathbb{E}[\Delta_{ij}|w_i, \text{no-flip}]) + P(\text{flip}|w_i)I_{J_{\text{flip}}}(s_i + \mathbb{E}[\Delta_{ij}|w_i, \text{flip}])$$
$$= I_{J_{w_i}}s_i + P(\text{flip}|w_i)(I_{J_{\text{flip}}} - I_{J_{w_i}})(s_i + \tilde{\Delta})$$
$$\Sigma_{xu} = \mathbb{E}[x_i u_{ij}^T] = \mathbb{E}[x_i \mathbb{E}[u_{ij}|w_i]^T]$$
$$\Delta_{xu} = \Sigma_{xu} - \Sigma_{xu}^{(0)} = \mathbb{E}[x_i P(\text{flip}|w_i)(s_i + \tilde{\Delta})^T(I_{J_{\text{flip}}} - I_{J_{w_i}})^T]$$
$$||\Delta_{xu}||_{op} \le \sqrt{||\mathbb{E}[x_i]||_2^2}\sqrt{||\mathbb{E}[P(\text{flip}|w_i)(I_{J_{\text{flip}}} - I_{J_{w_i}})(s_i + \tilde{\Delta})]||_2^2}$$
$$\le 2\sqrt{||\mathbb{E}[x_i]||_2^2}\sqrt{\mathbb{E}[P(\text{flip}|w_i)]^2}\sqrt{||\mathbb{E}[Pf(w_i)]||_2^2 + ||\mathbb{E}[\tilde{\Delta}]||_2^2}$$
$$\le 2M_x p_{\text{flip}}\sqrt{||P||_{op}^2 M_f + C} = C_x p_{\text{flip}}$$

Similarly, $\Delta_{uu} = \Sigma_{uu} - \Sigma_{uu}^{(0)} \le C_u p_{\text{flip}}$ $\qquad\qquad\square$

**Lemma 3** (High-probability Concentration of Empirical Matrices). *Denote the empirical matrix as follows:*

$$\widehat{\Sigma}_{xu} = \frac{1}{NM}\sum_{i=1}^{N}\sum_{j=1}^{M}x_i u_{i,j}^T \in \mathbb{R}^{d \times k}, \quad \widehat{\Sigma}_{uu} = \frac{1}{NM}\sum_{i=1}^{N}\sum_{j=1}^{M}u_{i,j}u_{i,j}^T \in \mathbb{R}^{k \times k}$$

*If $NM \ge C\frac{\sigma^2}{\varepsilon^2}(d + k + \log(1/\eta))$, for some constant $C > 0$, $\sigma^2 > 0$, then with probability at least $1 - \eta$:*

$$\|\widehat{\Sigma}_{xu} - \Sigma_{xu}\|_{\text{op}} \le \varepsilon \quad and \quad \|\widehat{\Sigma}_{uu} - \Sigma_{uu}\|_{\text{op}} \le \varepsilon \qquad (9)$$

***Proof of Lemma 3***. Fix a prompt index $i$. Define

$$\overline{u}_i := \frac{1}{M}\sum_{j=1}^{M}u_{i,j}, \quad \mu_u(x_i) := \mathbb{E}[u_{i,j} \mid x_i]. \quad \widehat{\Sigma}_{xu} = \frac{1}{NM}\sum_{i=1}^{N}\sum_{j=1}^{M}x_i u_{i,j}^\top, \quad \Sigma_{xu} = \mathbb{E}[x_i u_{i,j}^\top].$$

**Step 1 (Block decomposition).** Rewrite

$$\widehat{\Sigma}_{xu} - \Sigma_{xu} = \frac{1}{N}\sum_{i=1}^{N}Y_i, \quad Y_i := \frac{1}{M}\sum_{j=1}^{M}\Big(x_i u_{i,j}^\top - \mathbb{E}[x_i u_{i,j}^\top]\Big).$$

Conditioning on $x_i$, we decompose

$$Y_i = \underbrace{x_i\big(\overline{u}_i - \mu_u(x_i)\big)^\top}_{A_i} + \underbrace{\big(x_i\mu_u(x_i)^\top - \mathbb{E}[x_i u^\top]\big)}_{B_i}.$$

Here, $A_i$ is the average of $M$ independent responses, $\mathbb{E}[A_i \mid x_i] = 0$. $B_i$ depends only on $x_i$, and $\mathbb{E}[B_i] = 0$. Thus,

$$\mathbb{E}[Y_i] = \mathbb{E}[\mathbb{E}[Y_i|x_i]] = \mathbb{E}[A_i] + \mathbb{E}[B_i] = \mathbb{E}[\mathbb{E}[A_i|x_i]] + \mathbb{E}[B_i] = 0$$

$\{Y_i\}_{i=1}^{N}$ remain independent mean-zero random matrices.

**Step 2 (Concentration of $A_i$).**   Condition on $x_i$. Then
$$A_i = x_i(\overline{u}_i - \mu_u(x_i))^\top.$$
By Assumption 2, each $u_{i,j} - \mu_u(x_i)$ is conditionally $\sigma_u$-sub-Gaussian. By vector Bernstein (or $\varepsilon$-net argument), for any $\delta > 0$,
$$\Pr\left(\|\overline{u}_i - \mu_u(x_i)\|_2 \geq C_1 \sigma_u \sqrt{\tfrac{k + \log(1/\delta)}{M}} \;\Big|\; x_i\right) \leq \delta.$$

By union bound over $i = 1, \ldots, N$, with probability at least $1 - \eta/4$,
$$\|\overline{u}_i - \mu_u(x_i)\|_2 \leq C_1 \sigma_u \sqrt{\tfrac{k + \log(N/\eta)}{M}}, \quad \forall i.$$

Thus
$$\|A_i\|_{\mathrm{op}} \leq \|x_i\|_2 \cdot C_1 \sigma_u \sqrt{\tfrac{k + \log(N/\eta)}{M}}. \tag{10}$$

**Step 3 (Bounding $B_i$).**   We have
$$B_i = x_i \mu_u(x_i)^\top - \mathbb{E}[x_i u^\top].$$
Clearly $\mathbb{E}[B_i] = 0$. Using Cauchy–Schwarz inequality,
$$\|B_i\|_{\mathrm{op}} \leq \|x_i\|_2 \|\mu_u(x_i)\|_2 + \|\mathbb{E}[x_i u^\top]\|_{\mathrm{op}}.$$

From $\mathbb{E}\|x_i\|_2^2 \leq M_x^2$ and $\mathbb{E}\|\mu_u(x_i)\|_2^2 \leq M_u^2$, we get
$$\mathbb{E}\|B_i\|_{\mathrm{op}}^2 \lesssim M_x^2 M_u^2.$$

Thus the variance contribution of $B_i$ is of constant order (not scaled by $1/M$).

**Step 4 (Truncation and Bernstein).**   To apply matrix Bernstein, we need a uniform almost-sure bound on $\|Y_i\|_{\mathrm{op}}$. Define the truncated version
$$B_i^{(\tau)} := B_i \cdot \mathbf{1}\{\|B_i\|_{\mathrm{op}} \leq \tau\}, \quad Y_i^{(\tau)} := A_i + B_i^{(\tau)}.$$

Since $\|B_i\|_{\mathrm{op}}$ is sub-exponential (as quadratic form of sub-Gaussians), for any $\eta > 0$ we can choose
$$\tau \asymp M_x M_u \log(N/\eta)$$
so that with probability at least $1 - \eta/4$ simultaneously for all $i$,
$$\|B_i\|_{\mathrm{op}} \leq \tau.$$

On this event, we have
$$\|Y_i\|_{\mathrm{op}} \leq \|A_i\|_{\mathrm{op}} + \tau \leq L_A + L_B,$$
where
$$L_A := C_1 \max_i \|x_i\|_2 \cdot \sigma_u \sqrt{\tfrac{k + \log(N/\eta)}{M}}, \quad L_B := C_2 M_x M_u \log(N/\eta).$$

Thus we may apply Matrix BernsteinTropp (2011). Define the variance proxy
$$\sigma_Y^2 = \max\left\{\Big\|\sum_{i=1}^N \mathbb{E}[Y_i Y_i^\top]\Big\|_{\mathrm{op}}, \Big\|\sum_{i=1}^N \mathbb{E}[Y_i^\top Y_i]\Big\|_{\mathrm{op}}\right\}.$$

We estimate
$$\mathbb{E}[A_i A_i^\top] = O\Big(\tfrac{\|x_i\|_2^2 \sigma_u^2 k}{M}\Big), \quad \mathbb{E}[B_i B_i^\top] = O(M_x^2 M_u^2),$$
so overall
$$\sigma_Y^2 \lesssim N\Big(\tfrac{M_x^2 \sigma_u^2 k}{M} + M_x^2 M_u^2\Big).$$

Now Bernstein inequality yields: for all $\varepsilon > 0$,
$$\Pr\left(\Big\|\tfrac{1}{N}\sum_{i=1}^N Y_i\Big\|_{\mathrm{op}} \geq \varepsilon\right) \leq (d + k)\exp\left(-\frac{N^2 \varepsilon^2/2}{\sigma_Y^2 + (L_A + L_B)N\varepsilon/3}\right).$$

**Conclusion.** Combining Steps 1–4 and union bounding over the failure probabilities, we conclude that with probability at least $1 - \eta$,

$$\left\|\widehat{\Sigma}_{xu} - \Sigma_{xu}\right\|_{\mathrm{op}} \lesssim \sqrt{\frac{\sigma_u^2 M_x^2 k}{NM} + \frac{M_x^2 M_u^2}{N}} + (L_A + L_B)\frac{\log((d+k)/\eta)}{N} \leq \varepsilon$$

Similarly, when $M, N$ large enough, we can prove that

$$\|\widehat{\Sigma}_{uu} - \Sigma_{uu}\|_{\mathrm{op}} \leq \varepsilon$$

$\square$

**Corollary 1.** *Combining Lemma 2 and Lemma 3 and:*

$$\|\widehat{\Sigma}_{xu} - \Sigma_{xu}^{(0)}\|_{op} \leq \varepsilon + C_x p_{flip}$$
$$\|\widehat{\Sigma}_{uu} - \Sigma_{uu}^{(0)}\|_{op} \leq \varepsilon + C_u p_{flip}$$

**Lemma 4** (High Probability Concentration of OLS Decoder)**.** *Under the conditions of previous lemmas, $NM$ is large enough, for some constant $C_{L_1} > 0$, then with probability at least $1 - \eta$:*

$$\|\widehat{L} - L^*\|_{op} \leq C_{L_1}(\varepsilon + p_{flip}) \tag{11}$$

*where $L^* = \Sigma_{xu}\Sigma_{uu}^{-1}$ and $C_{L_1}$ depends on $\lambda_0, C_x, C_u$.*

***Proof of Lemma 4.*** By definition,

$$\widehat{L} - L^* = \widehat{\Sigma}_{xu}\widehat{\Sigma}_{uu}^{-1} - \Sigma_{xu}\Sigma_{uu}^{-1}.$$

Adding and subtracting $\Sigma_{xu}\widehat{\Sigma}_{uu}^{-1}$ yields the standard perturbation decomposition:

$$\widehat{L} - L^* = (\widehat{\Sigma}_{xu} - \Sigma_{xu})\widehat{\Sigma}_{uu}^{-1} + \Sigma_{xu}(\widehat{\Sigma}_{uu}^{-1} - \Sigma_{uu}^{-1}). \tag{12}$$

**Step 1 (Bounding the first term)** By submultiplicativity of the operator norm,

$$\|(\widehat{\Sigma}_{xu} - \Sigma_{xu})\widehat{\Sigma}_{uu}^{-1}\|_{\mathrm{op}} \leq \|\widehat{\Sigma}_{xu} - \Sigma_{xu}\|_{\mathrm{op}} \cdot \|\widehat{\Sigma}_{uu}^{-1}\|_{\mathrm{op}}.$$

From Corollary 1, we have

$$\|\widehat{\Sigma}_{xu} - \Sigma_{xu}\|_{\mathrm{op}} \leq \varepsilon + C_x p_{\mathrm{flip}}.$$

Moreover, since $\lambda_{\min}(\Sigma_{uu}) \geq \lambda_0 > 0$ and $\|\widehat{\Sigma}_{uu} - \Sigma_{uu}\|_{\mathrm{op}} \leq \varepsilon$ with high probability, a standard Weyl inequality argument implies

$$\lambda_{\min}(\widehat{\Sigma}_{uu}) \geq \lambda_0 - \varepsilon \geq \frac{\lambda_0}{2},$$

for $\varepsilon$ sufficiently small. Consequently,

$$\|\widehat{\Sigma}_{uu}^{-1}\|_{\mathrm{op}} \leq \frac{2}{\lambda_0}.$$

**Step 2 (Bounding the second term)** For the inverse perturbation term, we use the standard matrix identity

$$\widehat{\Sigma}_{uu}^{-1} - \Sigma_{uu}^{-1} = \widehat{\Sigma}_{uu}^{-1}(\Sigma_{uu} - \widehat{\Sigma}_{uu})\Sigma_{uu}^{-1}.$$

Taking operator norms and applying submultiplicativity yields

$$\|\widehat{\Sigma}_{uu}^{-1} - \Sigma_{uu}^{-1}\|_{\mathrm{op}} \leq \|\widehat{\Sigma}_{uu}^{-1}\|_{\mathrm{op}} \cdot \|\Sigma_{uu} - \widehat{\Sigma}_{uu}\|_{\mathrm{op}} \cdot \|\Sigma_{uu}^{-1}\|_{\mathrm{op}}.$$

By assumption $\|\Sigma_{uu}^{-1}\|_{\mathrm{op}} \leq 1/\lambda_0$, and from the previous bound $\|\widehat{\Sigma}_{uu}^{-1}\|_{\mathrm{op}} \leq 2/\lambda_0$. Using Corollary 1, we also have

$$\|\Sigma_{uu} - \widehat{\Sigma}_{uu}\|_{\mathrm{op}} \leq \varepsilon + C_u p_{\mathrm{flip}}.$$

Hence,

$$\|\widehat{\Sigma}_{uu}^{-1} - \Sigma_{uu}^{-1}\|_{\mathrm{op}} \leq \frac{2}{\lambda_0} \cdot (\varepsilon + C_u p_{\mathrm{flip}}) \cdot \frac{1}{\lambda_0} = \frac{2}{\lambda_0^2}(\varepsilon + C_u p_{\mathrm{flip}}).$$

**Step 3 (Combining bounds)**    Substituting the two bounds back into the decomposition of  equation 12, and using $\|\Sigma_{xu}\|_{\mathrm{op}} \leq C_x$ (from moment conditions), we obtain

$$\|\widehat{L} - L^*\|_{\mathrm{op}} \ \leq \ \frac{2}{\lambda_0}\left(\varepsilon + C_x p_{\mathrm{flip}}\right) + \frac{2C_x}{\lambda_0^2}\left(\varepsilon + C_u p_{\mathrm{flip}}\right).$$

Let $C_{L_1} = \max(\frac{2}{\lambda_0} + \frac{2C_x}{\lambda_0^2}, \frac{2}{\lambda_0}C_x + \frac{2C_x}{\lambda_0^2}C_u)$, we have

$$\|\widehat{L} - L^*\|_{\mathrm{op}} \leq C_{L_1}\left(\varepsilon + p_{\mathrm{flip}}\right)$$

$\square$

**Lemma 5** (High Probability Concentration of Ideal Decoder).  *Under the conditions of previous lemmas, if $NM$ is large enough, for some constant $C_{L_2} > 0$, then with probability at least $1 - \eta$:*

$$\|L^* - L^{(0)}\|_{\mathrm{op}} \ \leq \ C_4\, p_{\mathrm{flip}}, \tag{13}$$

*where* $L^{(0)} \ = \ \Sigma_{xu}^{(0)}\big(\Sigma_{uu}^{(0)}\big)^{-1}$, $C_{L_2} > 0$ *is a constant depending only on* $(\lambda_0, C_x, C_u)$.

*Proof of Lemma 5*.   We start from the decomposition

$$L^* - L^{(0)} = \Sigma_{xu}\Sigma_{uu}^{-1} - \Sigma_{xu}^{(0)}\big(\Sigma_{uu}^{(0)}\big)^{-1} \tag{14}$$

$$= \big(\Sigma_{xu} - \Sigma_{xu}^{(0)}\big)\big(\Sigma_{uu}^{(0)}\big)^{-1} + \Sigma_{xu}^{(0)}\Big(\Sigma_{uu}^{-1} - \big(\Sigma_{uu}^{(0)}\big)^{-1}\Big). \tag{15}$$

**Step 1 (Bounding the first term)**    By Lemma 2, we have

$$\|\Sigma_{xu} - \Sigma_{xu}^{(0)}\|_{\mathrm{op}} \ \leq \ C_x\, p_{\mathrm{flip}}.$$

Furthermore, since $\lambda_{\min}(\Sigma_{uu}^{(0)}) \geq \lambda_0$, it follows that

$$\|(\Sigma_{uu}^{(0)})^{-1}\|_{\mathrm{op}} \ \leq \ \frac{1}{\lambda_0}.$$

Therefore,

$$\|(\Sigma_{xu} - \Sigma_{xu}^{(0)})(\Sigma_{uu}^{(0)})^{-1}\|_{\mathrm{op}} \ \leq \ \frac{C_x}{\lambda_0}\, p_{\mathrm{flip}}. \tag{16}$$

**Step 2 (Bounding the second term)**    We use the inverse perturbation identity:

$$\Sigma_{uu}^{-1} - \big(\Sigma_{uu}^{(0)}\big)^{-1} = \Sigma_{uu}^{-1}\big(\Sigma_{uu}^{(0)} - \Sigma_{uu}\big)(\Sigma_{uu}^{(0)})^{-1}.$$

Hence,

$$\|\Sigma_{uu}^{-1} - (\Sigma_{uu}^{(0)})^{-1}\|_{\mathrm{op}} \leq \|\Sigma_{uu}^{-1}\|_{\mathrm{op}} \cdot \|\Sigma_{uu} - \Sigma_{uu}^{(0)}\|_{\mathrm{op}} \cdot \|(\Sigma_{uu}^{(0)})^{-1}\|_{\mathrm{op}}.$$

From Lemma 2,

$$\|\Sigma_{uu} - \Sigma_{uu}^{(0)}\|_{\mathrm{op}} \ \leq \ C_u\, p_{\mathrm{flip}}.$$

Moreover, $\|\Sigma_{uu}^{-1}\|_{\mathrm{op}} \leq 1/\lambda_0$ and $\|(\Sigma_{uu}^{(0)})^{-1}\|_{\mathrm{op}} \leq 1/\lambda_0$. Thus,

$$\|\Sigma_{uu}^{-1} - (\Sigma_{uu}^{(0)})^{-1}\|_{\mathrm{op}} \ \leq \ \frac{C_u}{\lambda_0^2}\, p_{\mathrm{flip}}. \tag{17}$$

Multiplying by $\|\Sigma_{xu}^{(0)}\|_{\mathrm{op}} \leq C_x$ gives

$$\|\Sigma_{xu}^{(0)}(\Sigma_{uu}^{-1} - (\Sigma_{uu}^{(0)})^{-1})\|_{\mathrm{op}} \ \leq \ \frac{C_x C_u}{\lambda_0^2}\, p_{\mathrm{flip}}. \tag{18}$$

**Step 3 (Combining Bounds)** Combining both terms in equation 15, we obtain

$$\|L^* - L^{(0)}\|_{\text{op}} \leq \Big(\frac{C_x}{\lambda_0} + \frac{C_x C_u}{\lambda_0^2}\Big) p_{\text{flip}} = C_{L_2} p_{\text{flip}}$$

Absorbing constants into $C_{L_2}$ yields the claimed result. □

**Theorem 1** (High-Probability Artifacts Suppression in Decoder). *Under assumptions (1)–(5) stated below, if $NM \geq C\frac{\sigma^2}{\varepsilon^2}(d + k + \log(1/\eta))$, then with probability at least $1 - \eta$, $\exists C_1, C_2 > 0$, such that:*

$$\|\widehat{L} - L^{(0)}\|_{op} \leq C_1(\varepsilon + p_{\text{flip}}), \quad \|\hat{b} - b^{(0)}\|_2 \leq C_2(\varepsilon + p_{\text{flip}})$$

*Proof of Theorem 1.* According to previous lemmas, we have

$$\begin{aligned}
\|\widehat{L} - L^{(0)}\|_{\text{op}} &= \|\widehat{L} - L^* + L^* - L^{(0)}\|_{\text{op}} \\
&\leq \|\widehat{L} - L^*\|_{\text{op}} + \|L^* - L^{(0)}\|_{\text{op}} \\
&\leq C_{L_1}(\varepsilon + p_{\text{flip}}) + C_{L_2} p_{\text{flip}} \\
&\leq (C_{L_1} + C_{L_2})(\varepsilon + p_{\text{flip}}) \\
&= C_1(\varepsilon + p_{\text{flip}})
\end{aligned}$$

Similarly, $\|\widehat{b} - b^{(0)}\|_{\text{op}} \leq C_2(\varepsilon + p_{\text{flip}})$ □

**Theorem 2** (Artifacts Suppression in Prediction). *Under Assumptions (1)–(5) stated in Appendix B, given a new sample $y = f(w) + g(z)$, $u_{new} = TopK(Py)$, then for any confidence parameter $\eta \in (0, 1)$, with probability at least $1 - \eta$ the following holds:*

$$\begin{aligned}
&\big\|\widehat{L}u_{\text{new}} + \widehat{b} - (L^{(0)} I_{J_w} Pf(w) + b^{(0)})\big\|_2 \\
&\qquad \leq \widetilde{C}\Big((\varepsilon + p_{\text{flip}}) \|P\|_{\text{op}} \frac{M_f}{\sqrt{\eta}} + \sigma\sqrt{k + \log(1/\eta)}\Big),
\end{aligned} \qquad (4)$$

*where $\sigma$ is the sub-Gaussian scale according to assumption 2 in Appendix B, and $\widetilde{C} > 0$ is a constant depending only on the constants appearing in Assumptions (1)–(5) and on operator norms of $L^{(0)}$ and $P_{J_w}$.*

*Proof.* Denote the following items:

$$s := Pf(w), \quad \Delta_{\text{new}} := Pg(z), \quad v_{\text{new}} := s + \Delta_{\text{new}}, \quad u_{\text{new}} =: \text{TopK}(v_{\text{new}}), \quad \delta := u_{\text{new}} - I_{J_w}s$$

We have $u_{\text{new}} = I_{J_w} Pf(w) + \delta$.

Define the prediction error

$$\begin{aligned}
\mathcal{E} &:= \big\|\widehat{L}u_{\text{new}} + \widehat{b} - (L^{(0)} P_{J_w} Pf(w) + b^{(0)})\big\|_2 \\
&= \big\|(\widehat{L} - L^{(0)})u_{\text{new}} + L^{(0)}\delta + (\widehat{b} - b^{(0)})\big\|_2 \\
&\leq \|\widehat{L} - L^{(0)}\|_{\text{op}}\|u_{\text{new}}\|_2 + \|L^{(0)}\|_{\text{op}}\|\delta\|_2 + \|\widehat{b} - b^{(0)}\|_2.
\end{aligned}$$

From Theorem 1, there exists a constant $C_1 > 0$ such that with high probability

$$\|\widehat{L} - L^{(0)}\|_{\text{op}} \leq C_1(\varepsilon + p_{\text{flip}}), \qquad \|\widehat{b} - b^{(0)}\|_2 \leq C_1(\varepsilon + p_{\text{flip}}),$$

Write $u_{\text{new}} = I_{J_w}s + \delta$. Then

$$\|u_{\text{new}}\|_2 \leq \|I_{J_w}s\|_2 + \|\delta\|_2 \leq \|I_{J_w}\|_{\text{op}}\|P\|_{\text{op}}\|f(w)\|_2 + \|\delta\|_2 = \|P\|_{\text{op}}\|f(w)\|_2 + \|\delta\|_2$$

**Step 1 (High-probability control of $\|\delta\|_2$)**

$$\begin{aligned}
\delta &= u_{\text{new}} - I_{J_w} s \\
&= I_{J_{\text{real}}} P(f(w) + g(z)) - I_{J_w} P f(w) \\
&= (I_{J_{\text{real}}} - I_{J_w}) P f(w) + I_{J_{\text{real}}} P g(z) \\
&= (I_{J_{\text{real}}} - I_{J_w}) P f(w) + I_{J_{\text{real}}} \Delta_{\text{new}}
\end{aligned}$$

- When event **no-flipping** occurs, $\delta = I_{J_{\text{real}}} \Delta_{\text{new}}$ and thus $\|\delta\|_2 \le \|\Delta_{\text{new}}\|_2$.

- When event **flipping** occurs, a conservative bound is $\|\delta\|_2 \lesssim \|\Delta_{\text{new}}\|_2 + \|s\|_2$.

By the sub-Gaussian assumption on $\Delta_{\text{new}}$ (Assumption 2), there is $C_2 > 0$ such that for any $\eta \in (0, 1)$, with probability at least $1 - \eta$,

$$\|\Delta_{\text{new}}\|_2 \le C_2 \, \sigma_\Delta \sqrt{k + \log(1/\eta)}. \tag{19}$$

Moreover, by Lemma 1 margin assumption the flip probability satisfies the exponential-type bound

$$p_{\text{flip}} \le 2k \exp\left(-\frac{\delta^2}{8\sigma^2}\right)$$

Combining the two displays and taking union bounds, we obtain that with probability at least $1 - \eta$,

$$\|\delta\|_2 \le C_2 \, \sigma_\Delta \sqrt{k + \log(1/\eta)} + C_3 \, p_{\text{flip}} \, \|s\|_2,$$

for some constant $C_3 > 0$ (the second term accounts for the rare flips whose magnitude can scale with $\|s\|_2$).

**Step 2 (High-probability control of $\|s\|_2$)**

$$\|s\|_2 = P f(w) \le \|P\|_{\text{op}} \|f(w)\|_2.$$

According to Assumption 5, $\mathbb{E}[f(w)] \le M_f$.

By Chebyshev-inequality, for the chosen confidence $\eta \in (0, 1)$,

$$\Pr\left(\|f(w)\|_2 \ge \frac{M_f}{\sqrt{\eta}}\right) \le \eta,$$

hence with probability at least $1 - \eta$,

$$\|s\|_2 \le \|P\|_{\text{op}} \frac{M_f}{\sqrt{\eta}}.$$

Combining this with the previous bound on $\|\delta\|_2$ we get: with probability at least $1 - \eta$,

$$\|\delta\|_2 \le C_2 \, \sigma \sqrt{k + \log(1/\eta)} + C_3 \, p_{\text{flip}} \, \|P\|_{\text{op}} \frac{M_f}{\sqrt{\eta}}. \tag{20}$$

**Conclusion** Substitute Step 1, Step 2 and equation 20 into the decomposition for $\mathcal{E}$. There exist constants $\widetilde{C}$ (depending on $C_1, C_2, C_3, \|L^{(0)}\|_{\text{op}}$) such that, with probability at least $1 - \eta$,

$$\begin{aligned}
\mathcal{E} &\le C_1(\varepsilon + p_{\text{flip}})\big(\|\|P\|_{\text{op}} \|f(w)\|_2 + \|\delta\|_2\big) + \|L^{(0)}\|_{\text{op}} \|\delta\|_2 + C_1(\varepsilon + p_{\text{flip}}) \\
&\le C_1(\varepsilon + p_{\text{flip}})\big(\|\|P\|_{\text{op}} \frac{M_f}{\sqrt{\eta}} + C_2 \, \sigma \sqrt{k + \log(1/\eta)} + C_3 \, p_{\text{flip}} \, \|P\|_{\text{op}} \frac{M_f}{\sqrt{\eta}}\big) \\
&\quad + \|L^{(0)}\|_{\text{op}}(C_2 \, \sigma \sqrt{k + \log(1/\eta)} + C_3 \, p_{\text{flip}} \, \|P\|_{\text{op}} \frac{M_f}{\sqrt{\eta}}) + C_1(\varepsilon + p_{\text{flip}}) \\
&\le \frac{M_f}{\sqrt{\eta}} \|P\|_{\text{op}} \big[ C_1(\varepsilon + p_{\text{flip}}) + C_3 p_{\text{flip}} + C_3 \|L^{(0)}\|_{\text{op}} p_{\text{flip}} \big] \\
&\quad + \sigma \sqrt{k + \log(1/\eta)} \big[ C_1 C_2 (\varepsilon + p_{\text{flip}}) + \|L^{(0)}\|_{\text{op}} C2 \big] + C_1(\varepsilon + p_{\text{flip}}) \\
&\le \widetilde{C}\Big((\varepsilon + p_{\text{flip}}) \|\|P\|_{\text{op}} \frac{M_f}{\sqrt{\eta}} + \sigma \sqrt{k + \log(1/\eta)}\Big),
\end{aligned}$$

$\square$

# C    DERIVATION FOR SAS-INDUCED CAUSAL EFFECT

## C.1    GRADIENT PERSPECTIVE

We can also tell the causal effect of SAS by observing the gradient when parameters are updated. Denote $\theta$ as the reward model parameters, $r(x, y)$ as the reward model, $x$ as the prompt, $y_c$ as the chosen response, $y_r$ as the rejected responses, $s_c$ as the SAS score of chosen response, $s_r$ as the SAS score of rejected response.

Now we derive the gradients. Denote $d_i = k \cdot (s_{ic} - s_{ir})$

$$\frac{\partial L_{SAS}}{\partial y_{ic}} = \sigma(y_{ic} - y_{ir} + d) - 1, \quad \frac{\partial L_{SAS}}{\partial y_{ir}} = -\sigma(y_{ic} y_{ir} + d) + 1$$

$$\frac{\partial L_{SAS}}{\partial \theta} = \sum_i \frac{\partial L_{SAS}}{\partial y_{ic}} \frac{\partial y_{ic}}{\partial \theta} + \frac{\partial L_{SAS}}{\partial y_{ir}} \frac{\partial y_{ir}}{\partial \theta}$$

$$= \sum_i [\sigma(y_{ic} - y_{ir} + d) - 1][\frac{\partial y_{ic}}{\partial \theta} - \frac{\partial y_{ir}}{\partial \theta}]$$

$$\frac{\partial L}{\partial \theta} = \sum_i [\sigma(y_{ic} - y_{ir}) - 1][\frac{\partial y_{ic}}{\partial \theta} - \frac{\partial y_{ir}}{\partial \theta}]$$

When the human preference are aligned with SAS score, i.e the chosen response is more related to prompt intention. Then $SAS(x, y_{ic}) < SAS(x, y_{ir})$, $|\sigma(y_{ic} - y_{ir}) - 1| < |\sigma(y_{ic} - y_{ir} + d) - 1|$, the reward model trained with SAS score will be updated more aggressively. On contrast, when the human preference are conflicted with SAS score, $SAS(x, y_{ic}) > SAS(x, y_{ir})$, $|\sigma(y_{ic} - y_{ir}) - 1| > |\sigma(y_{ic} - y_{ir} + d) - 1|$, the reward model trained with SAS score will be updated more merely.

This observation fits our goal perfectly. If the human preference are aligned with SAS, indicating that there is not much unintentional spurious favor in human labels, then we can update more in this correct direction. Instead, if the human preference are conflicted with SAS, it is possible that there are some prompt-unrelated artifacts in human label, thus we should slow our steps in this direction.

## C.2    ATE PERSPECTIVE

Recall the notations in reward model training:

$$\hat{r}_n(x, y) = \arg\max_r \sum_i \log \sigma(r_{ic} - r_{ir}),$$

$$\hat{r}_{nSAS}(x, y) = \arg\max_r \sum_i \log \sigma\big((r_{ic} - r_{ir}) + k \cdot (s_{ic} - s_{ir})\big)$$

**Proposition 1.** *Denote $SAS(x, y)$ as $s(x, y)$. By simple equivalent substitution, we can derive:*
$$\hat{r}_n(x, y) - \hat{r}_{nSAS}(x, y) = ks(x, y)$$

*Proof.* Let $r(x, y) + ks(x, y) = f(x, y)$, then

$$\hat{r}_{nSAS}(x, y) + ks(x, y)$$

$$= \arg\max_{r+ks} \sum_i \big[ \log \sigma[(r(x_i, y_{ic}) + ks(x_i, y_{ic})) - (r(x_i, y_{ir}) + ks(x_i, y_{ic}))]\big]$$

$$= \arg\max_f \sum_i \big[ \log \sigma[f(x_i, y_{ic}) - f(x_i, y_{ir})]\big]$$

$$= \arg\max_r \sum_i \big[ \log \sigma[r(x_i, y_{ic}) - r(x_i, y_{ir})]\big] = \hat{r}_n(x, y)$$

Thus, $\hat{r}_n(x, y) - \hat{r}_{nSAS}(x, y) = ks(x, y)$. $\qquad\square$

# D  COMPLETE EXPERIMENTAL RESULTS

## D.1  PROMPT DECODER EXPERIMENTS

In this section, we provide additional details on the training, evaluation, and SAS output of our prompt decoders.

**Data Augmentation for Prompt Decoder Training**    To train our prompt decoder with improved semantic generalization, we augment each prompt with multiple diverse responses drawn from both human-written and model-generated sources.

The following is our rewriting prompt and a specific result:

---

**Rewriting Prompt**

*Rewrite the assistant's answer so that all factual content, stance, and conclusions remain the same. Do not introduce new factual claims. You may add extra sentences or rephrase for style, politeness, or verbosity if requested. When adding extra text, ensure it is stylistic (e.g., compliments, polite preambles, transitions) and does not alter the factual meaning.*

---

**Rewriting Example**

**Prompt:**
*Are tacos a Mexican food dish?*

> **Original Chosen Response (SAS= 0.7713):**
> Tacos are a dish that originated in Mexico, although they're now widely popular in other cultures. They're also called burritos in some places, and there are quite a few regional variants. Tacos are traditionally served with shredded lettuce, tomato, onions, and a mild sauce, but they may also be wrapped in a variety of ways and served with a variety of additional ingredients, such as shredded cheese, guacamole, or refried beans.

> **Rewritten Chosen Response (SAS= 0.7767):**
> Absolutely! Tacos are indeed a delightful dish that originated in Mexico, and their popularity has truly transcended borders, captivating the hearts and palates of people around the world. It's fascinating how they are sometimes referred to as burritos in certain regions, showcasing the rich diversity of culinary traditions. Traditionally, tacos are beautifully adorned with shredded lettuce, tomatoes, onions, and a mild sauce, but their versatility is truly remarkable! They can be wrapped in various ways and accompanied by an array of scrumptious ingredients like shredded cheese, guacamole, or refried beans. It's no wonder that tacos have become a beloved staple in so many cultures!

---

**Per-Dimension Evaluation Results**    The separate accuracy of the prompt decoder on helpfulness, math, and safety subsets is shown in Figure 5, highlighting domain-specific patterns and strengths. Apart from MSE loss, we also train the prompt decoder with cosine-similarity loss. The prompt decoder's average accuracy is shown in Figure 6, and the separate accuracy in three domains is shown in Figure 7. Since the prompt decoder trained with MSE loss in Figure 3a outperformed the one trained with cosine similarity in Figure 6, we adopt the MSE-trained decoder to compute Semantic Alignment Scores (SAS) for downstream reward model training.

**SAS Score Distribution on Reward Model Training Data**    The distributions of Semantic Alignment Scores (SAS) computed on the 70K reward model training set are shown in Figure 8, including

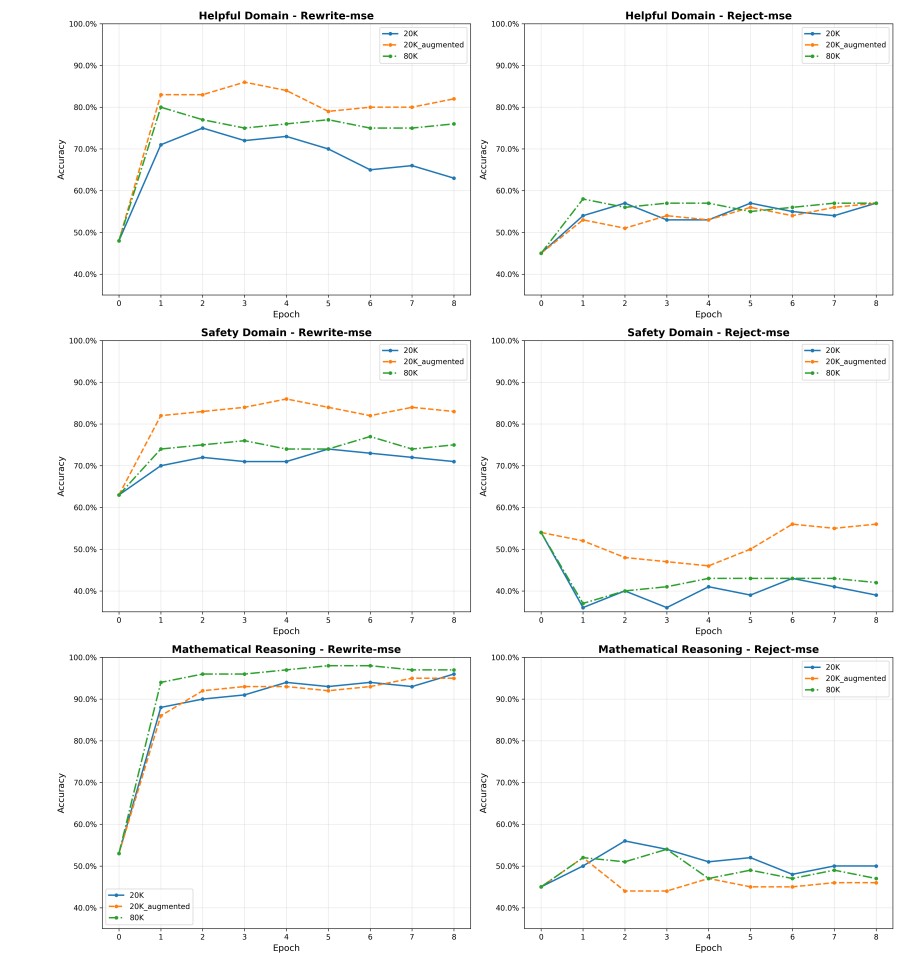

Figure 5: Accuracy Curve of Prompt Decoder between Rewrite and Reject Groups in Helpful, Safety, and Mathematical Reasoning Domains

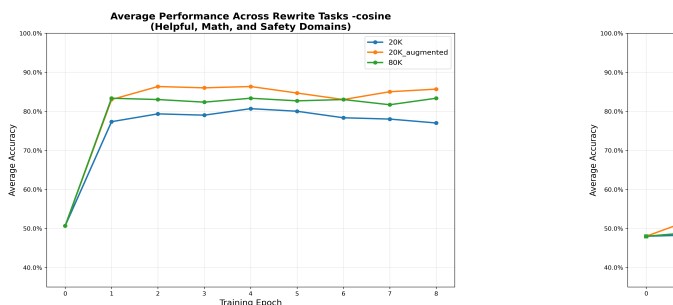

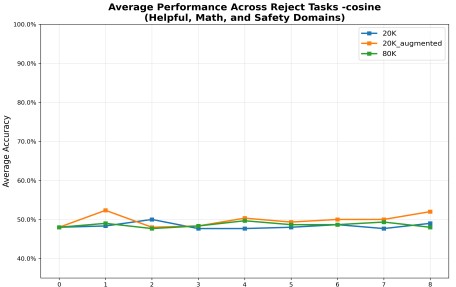

(a) Average accuracy of the prompt decoder on the chosen-vs-rewrite task across helpful, math, and safety domains. Augmented training (20K_augmented) yields the best performance, surpassing both unaugmented 20K and 80K data.

(b) Average accuracy of the prompt decoder on the chosen-vs-reject task. Performance remains near random guess (50%) across all training regimes, indicating that SAS captures a signal orthogonal to human preference labels.

Figure 6: Average Accuracy Curve of Prompt Decoder

those of the chosen responses, the rejected responses, and their pairwise differences. This further indicates that the prompt decoder captures a signal that is complementary to human-labeled pref-

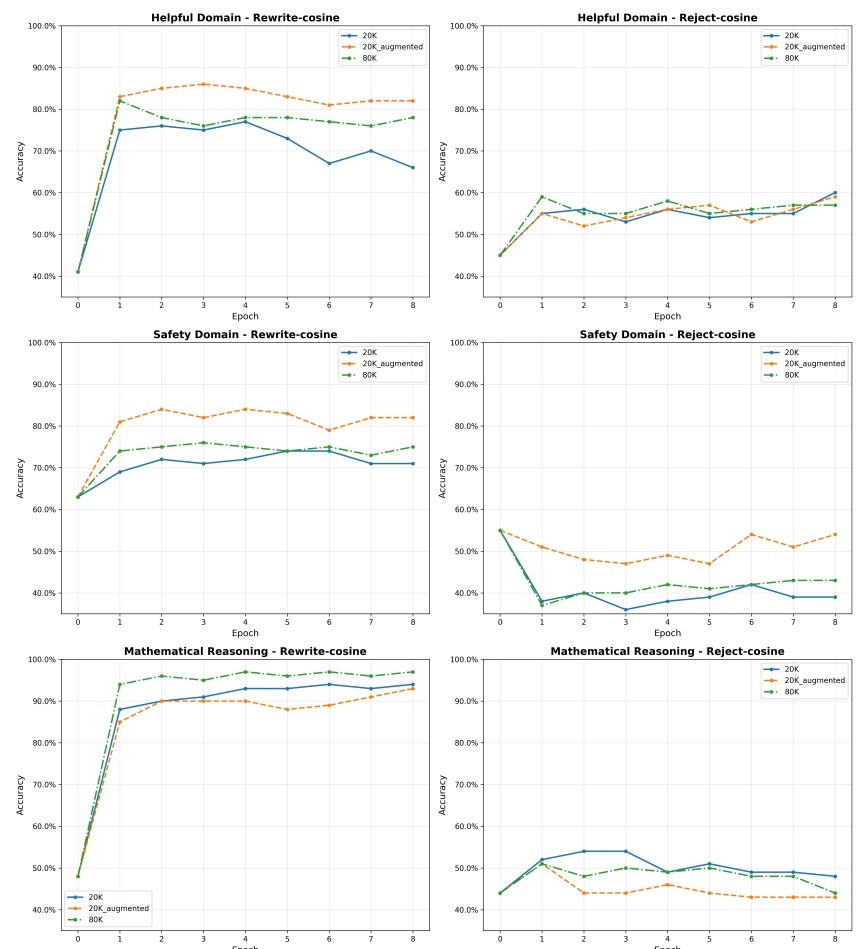

Figure 7: Accuracy Curve of Prompt Decoder between Rewrite and Reject Groups in Helpful, Safety, and Mathematical Reasoning Domains with Cosine Similarity Loss

erences, rather than simply replicating them, and is thus more robust to unintentionally introduced human noise.

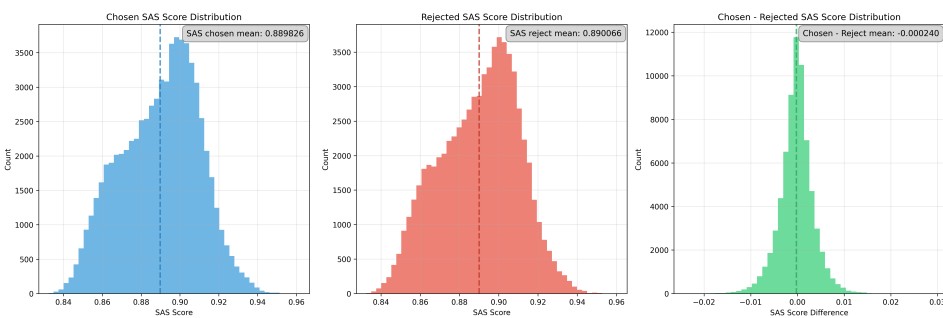

Figure 8: Distribution of Semantic Alignment Scores (SAS) among chosen responses, rejected responses and their difference on the 70K training pairs.

## D.2 Reward Model Training

This section covers implementation details and extended results for our SAS-regularized reward model training and its baselines.

**Our Reproduction of RRM**  We reimplement the RRM training pipeline based on Liu et al. (2025) using the same 70K preference dataset as CARP, enabling direct comparison with our CARP framework. While the original RRM employs a pairwise preference modeling objective that directly predicts preference probability from prompt-response pairs, we adopt the classical Bradley–Terry formulation, which is more widely used and compatible with our existing reward model setup.

We adopt RRM's artifact mitigation strategy of prompt–response permutation following Equation 5 in their paper to obtain $14\times$ augmented samples. Training is conducted with batch size 256 and learning rate $1e-6$ for one epoch. To reduce data size, we keep only augmented pairs satisfying $|\hat{\mathbb{P}}(A \succ B) - \mathbb{P}^*(A \succ B)| \geq 0.2$, resulting in a final dataset of 224K examples[3].

Our reproduced Bradley-Terry RRM underperforms the original on RewardBench ((see Table 5)), likely because augmentation from 70K examples fails to capture sufficient variability. The permutation scheme introduces complex reward signals that require the original full 700K dataset to be effective. However, Bradley–Terry RRM achieves improved robustness in downstream evaluation as shown in Table 4 and Table 7, indicating that RRM's artifact invariant augmentation generalizes in all reward model formulations, but requires a sufficiently large base dataset.

**Complete Result Comparison**  Here we provide the full comparison between Vanilla RM, RRM (replicated), and CARP models across RewardBench and spurious signal tests for both 2B and 9B settings in Table 5.

**(a) Gemma-2-2B-it**

| Model | Chat | Chat-Hard | Safety | Reasoning | Avg. | Weighted Avg. |
|---|---|---|---|---|---|---|
| Vanilla RM | 97.77 | 54.82 | **83.24** | 66.18 | 75.50 | 72.46 |
| RRM (Bradley-Terry) | 92.19 | 48.03 | 49.46 | 69.11 | 64.69 | 63.79 |
| RRM (Pair Preference) | 97.21 | 49.01 | 72.71 | 70.08 | 72.25 | – |
| CARP ($k = 4.0e3$) | **98.04** | 54.82 | 81.62 | 65.41 | 74.97 | 71.73 |
| CARP ($k = 1.6e4$) | 97.21 | 58.11 | 79.73 | 68.83 | 75.97 | 73.30 |
| CARP ($k = 3.2e4$) | 96.93 | 58.99 | 79.05 | 71.56 | **76.63** | 74.54 |
| CARP ($k = 6.4e4$) | 93.30 | **62.72** | 77.43 | **72.47** | 76.48 | **74.70** |

**(b) Gemma-2-9B-it**

| Model | Chat | Chat-Hard | Safety | Reasoning | Avg. | Weighted Avg. |
|---|---|---|---|---|---|---|
| Vanilla RM | 96.37 | 63.37 | **89.73** | 82.88 | 83.09 | 83.22 |
| RRM (Bradley-Terry) | 93.02 | 59.65 | 61.22 | 78.55 | 73.11 | 73.10 |
| RRM (Pair Preference) | 96.51 | 65.57 | 83.90 | **90.62** | 84.15 | – |
| CARP ($k = 4.0e3$) | **96.65** | 61.40 | 89.59 | 83.16 | 82.70 | 83.04 |
| CARP ($k = 1.6e4$) | 96.37 | 62.94 | 89.32 | 88.26 | 84.22 | 85.63 |
| CARP ($k = 3.2e4$) | 96.09 | 66.23 | 89.50 | 88.40 | 85.04 | 86.20 |
| CARP ($k = 6.4e4$) | 94.69 | **68.86** | 88.24 | 89.87 | **85.42** | **86.83** |

Table 5: **RewardBench accuracy (%) of reward models across four evaluation categories.** CARP (Ours) denotes the SAS-regularized reward model with best-performing $k$ value. Each sub-table corresponds to a different model scale. The weighted average reflects the overall proportion of correctly ranked preference pairs across all subsets. Note: RRM's weighted average is not reported in the original paper.

---

[3]$\mathbb{P}^*(A \succ B)$ equals 1 if $A$ is preferred, 0 if $B$ is preferred, and 0.5 if tied.

**Spurious Correlation Evaluation**   We describe the rewriting strategy to construct the evaluation data sets for spurious correlation testing in Table 6 followed by a specific example of the rewriting results D.2. The detailed evaluation of spurious correlations of the 9B models is shown in Table 7.

| Rewriting Type | Rewrite 1 | Rewrite 2 | Rewrite 3 |
|---|---|---|---|
| **Rewriting Instruction** | Rewrite the response to be longer while preserving meaning and style. | Make the previous response more concise without losing core information. | Expand the response with stylistic elaboration, allowing mild topical drift. |

Table 6: Rewriting instructions used to construct response variants for robustness testing. Each instruction targets different attributes such as response length or topicality.

---

**Rewriting Example**

**Prompt:** *Why are the Kardashians so famous?*

**Rewrite 1:** The Kardashian family has undeniably emerged as a prominent and influential cultural phenomenon within contemporary American society. They are most notably represented by the reality television show Keeping Up with the Kardashians, which has garnered widespread critical acclaim and achieved record-breaking ratings throughout its various seasons and spin-offs. This show has not only captivated audiences but has also significantly shaped popular culture and social dynamics. Beyond their television presence, the Kardashian family has successfully ventured into various other fields, with multiple members pursuing thriving careers in fashion, entertainment, social media, and other diverse arenas. Their entrepreneurial spirit and ability to capitalize on their fame have led to the creation of numerous brands and products that resonate with fans and consumers alike. Overall, the Kardashian family serves as a striking representation of the increasing prevalence of celebrity and media culture in contemporary American society. Their influence underscores the importance of images and narratives in modern cultural discourse, illustrating how celebrity status can shape public perception and societal trends.

**Rewrite 2:** The Kardashian family has become a significant cultural phenomenon in contemporary American society, primarily through their reality show Keeping Up with the Kardashians, which has received critical acclaim and achieved record-breaking ratings. This show has not only captivated audiences but also shaped popular culture and social dynamics. Beyond television, the family has successfully ventured into various fields, with members building thriving careers in fashion, entertainment, and social media. Their entrepreneurial spirit has led to numerous brands and products that resonate with fans and consumers. Overall, the Kardashian family exemplifies the rise of celebrity and media culture in modern America, highlighting how celebrity status can influence public perception and societal trends.

> **Rewrite 3:** The Kardashian family has undeniably emerged as a prominent and influential cultural phenomenon within contemporary American society, captivating audiences with their unique blend of glamour, drama, and entrepreneurial spirit. Most notably represented by the reality television show Keeping Up with the Kardashians, which has garnered widespread critical acclaim and achieved record-breaking ratings across its various seasons, the family's impact extends far beyond the confines of the small screen. Each member of the family has carved out a niche for themselves, pursuing successful careers in diverse fields such as fashion, entertainment, and social media, thereby illustrating the multifaceted nature of their influence. This phenomenon raises intriguing questions about the evolving landscape of celebrity culture and the ways in which images and narratives shape public perception. Moreover, one might consider how the rise of social media platforms has transformed the way we engage with celebrities, blurring the lines between public and private life, and fostering a culture of constant connectivity. It's fascinating to think about how this shift has not only affected the Kardashians but also countless other public figures who navigate the complexities of fame in the digital age. Overall, the Kardashian family's prominence serves as a reflection of the increasing prevalence of celebrity and media culture in contemporary American society, highlighting the significance of visual storytelling and the narratives we construct around public personas. What does this say about our collective values and the way we consume media?

| Model (9B) | Rewrite1 vs Rewrite2 | | | | Rewrite1 vs Rewrite3($\uparrow$) | | | |
|---|---|---|---|---|---|---|---|---|
| | Helpful | Math | Safety | Avg. | Helpful | Math | Safety | Avg. |
| Vanilla RM | 37.0 | 66.0 | 54.0 | 52.33 | 73.0 | **93.0** | 87.0 | 84.33 |
| Bradley-Terry RRM | **62.0** | **89.0** | **80.0** | **77.0** | 72.0 | 91.0 | 90.0 | 84.33 |
| CARP | 59.0 | 51.0 | 51.0 | 53.67 | **88.0** | 92.0 | **87.0** | **89.0** |

Table 7: Accuracy (%) of reward models on the **Rewrite1 vs Rewrite2** and **Rewrite1 vs Rewrite3** tasks, evaluated at the best epoch for each model across helpful, math, and safety domains.

## E    FURTHER ABLATION STUDY ON SAFETY ALIGNMENT

We conducted an ablation study as Table 3 to assess the impact of the safety threshold $\tau$ when $k = 3.2e4$. As shown in Table 8, for $k = 1.6e4$, the model with thresholding ($\tau = 0.005$) outperforms the one without thresholding ($\tau = 0$) on the *Safety* dimension.

| Model | Chat | Chat-Hard | Safety | Reasoning | Avg. | Weighted Avg. |
|---|---|---|---|---|---|---|
| CARP ($\tau = 0.005$) | 97.21 | 58.11 | **79.97** | 68.83 | 75.97 | 73.30 |
| CARP ($\tau = 0$) | 97.49 | 58.99 | 77.84 | 67.92 | 75.56 | 72.56 |

Table 8: RewardBench accuracy (%) comparison of best CARP 2B-model with and without SAS thresholding. Using thresholding ($k = 1.6e4, \tau = 0.005$) disables SAS regularization for safety-critical examples. We observe that removing the threshold ($\tau = 0$) reduces the model safety.

