# OpenReview forum: "CARP: Causal Alignment of Reward Models via Response-to-Prompt Prediction"
_ICLR.cc/2026/Conference — ICLR 2026 Conference Withdrawn Submission_

### Official Review · Reviewer_hxKw · 2025-10-26

**Soundness:** 3
**Presentation:** 2
**Contribution:** 2
**Rating:** 4
**Confidence:** 3

**Summary:**

This submission proposes CARP, a framework for causal alignment of RMs. The key idea is to measure how well a response reflects the latent prompt intention. It is tackles through a Semantic Alignment Score (SAS), computed by training a prompt decoder that reconstructs the original prompt embedding from a response representation. The reconstruction error is then used to regularize reward model training.

The authors provides some theoretical analysis to show that SAS is primarily dependent on the prompt intention variable (W) and suppresses context-independent artifacts (Z). Empirically, they demonstrate moderate improvements (≈3–4% RewardBench accuracy gain) for Gemma-2B-it and Gemma-9B-it models when compared with some baselines, and a reduction in spurious reward correlations (length, sycophancy).

**Strengths:**

There are several strengths and interesting technical contributions:

1. The causal formulation looks novel to the reviewer. The causal DAG formulation looks to be a meaningful conceptual advance over prior work that mitigate artifacts via data augmentation but don’t model prompt intention explicitly.

2. The authors provide theoretical section (Sec. 2.3), which provide a formal justification that the decoder output approximates independence from artifacts. The idea of using a Top-K sparse autoencoder for invariance is well-grounded in representation learning theory.

3. Incorporating SAS as a regularization term within the Bradley–Terry loss yields a simple interface to existing RLHF pipelines, given SAS estimates.

4. Some technical details look thoughtful. For example, the explicit thresholding mechanism to disable SAS when topical deviation indicates safer behavior is a practically relevant detail.

**Weaknesses:**

However, the paper may have considerate rooms for improvements, especially centered around experimentation and theoretical justifications. Below are some weaknesses of the submission:

1. Some assumptions in the theoretical analysis look strong. For example, assumptions like additivity y=f(w)+g(z) and Top-K index stability look quite strong to the reviewer and rarely verifiable in real RM embeddings. The proofs rely on independence between prompt intention and artifacts, which may not hold in realistic data where spurious correlations are entangled.

2. Complexity. The framework requires training a non-trivial prompt decoder and sparse autoencoder, increasing engineering and computational complexity relative to simpler data augmentation from existing work. The paper doesn’t quantify this additional cost or show whether SAS is robust when applied to larger models.

3. Experimental validation is weak. Only Gemma-2B and 9B models are tested; these are relatively small, and improvements are modest (≈3 pp) on RewardBench. The paper lacks SOTA comparison, which limits the ICLR-level competitiveness. Also, in Table 2, RRM somehow performs even worse than vanilla RM, which raises suspicion about reproducibility or hyperparameter alignment. This needs discussion.

4. Although the authors promise open release, the pipeline involves custom data (rewriting prompts, SFT corpora mixing, SAE training). It would help to explicitly list hyperparameters, decoder architecture, and HuggingFace links. Given the paper’s focus on causal interpretability, stronger evidence of reproducibility is essential for acceptance at ICLR from the reviewer's view.

**Questions:**

Questions: Please refer to reviews above. Here are some additional ones:

Why does RRM underperform vanilla RM (Table 2)?

Have you tried larger base models to verify scalability?

Are RewardBench gains statistically significant (e.g., bootstrapped CIs)?

Include exact dataset splits, decoder architecture, and pretrained SAE checkpoint identifiers.

Suggestions:

Fig. 3(b) is visually unclear (text size and zoom).

---

### Official Review · Reviewer_edEj · 2025-10-31

**Soundness:** 2
**Presentation:** 2
**Contribution:** 2
**Rating:** 2
**Confidence:** 4

**Summary:**

The paper proposes a new loss component for the Bradley-Terry reward model training that tries to capture the human intention behind a user prompt, which an ideal response should ideally identify. This component is Semantic Alignment Score (SAS), calculated as the reconstruction error of a specific prompt decoder trained using an external LLM's internal activations of the prompt. Theoretical analyses demonstrate that by incorporating SAS in reward model training, a causal link is built from the "prompt intentnion" to the reward signal. Experiments on 2 reward models show some performance improvements.

**Strengths:**

- The idea behind the paper is interesting and seems promising, identifying prompt intentions for learning the reward model.
- The theoretical analyses demonstrate that, in the defined scope (assuming the proposed ways of training the prompt decoder and calculating SAS work well), a causal link is established to incorporate the prompt intentions.

**Weaknesses:**

- Clarity: I find the quality of paper presentation, especially those related to mathematical notations, not satisfactory:
  - Line 97: C is not defined anywhere, and it is not in the graph. Did you mean A?
  - Line 131 - 135: many notations are used without definitions: $d_x$, $x_i$, $d$, $f$, $g$, $P_y$, $k$, *Encoder/Decoder* (are these functions? What are their domains?), $L$, $b$. Lack of explicit discussions makes this paper less readable.
  - Another important point about citation style: I am confused why most papers are cited without mentioning their published venues? For example, line 521 is accepted at ICML, and 524 at NeurIPS. Line 42: citation typo?
  - RRM should be introduced in more detail before being used as a baseline.
- Methodology: What about scenarios where an answer could be associated for many questions? This could be an important aspect to consider but is not covered by the current setup.
- Methodology and Experiments: The proposed pipeline has many varying factors which can affect the end performance: (1) the choice of the external LLM providing the ground-truth prompt embeddings, (2) the position where such embeddings (activations) are taken from the LLM, (3) the training process of the prompt decoder model, (4) the trade-off parameter in SAS training loss for reward model. It seems that the paper only included 1 choice for (1) and (2), and the sensitivity to them is not evaluated.
- Experiments: only two reward models are tested, and this does not seem enough. The standard deviations of the performance results are not reported. Given the small improvements to the vanilla reward model training, it is difficult to judge whether the empirical improvements by the proposed method are significant.

**Questions:**

See weaknesses above.
- Apart from Bradley-Terry reward models, there are also multi-regression reward models which do not rely on binary comparisons. Does your method apply to them as well?

---

### Official Review · Reviewer_f3Fy · 2025-11-01

**Soundness:** 2
**Presentation:** 3
**Contribution:** 3
**Rating:** 4
**Confidence:** 4

**Summary:**

This paper introduces CARP, a method designed to create embeddings that ideally capture semantically important pieces of information in responses to prompts, while discarding less relevant information, and then these embeddings are used during reward model training to create better classifications of responses to prompts. They show strong results on standard reward modeling benchmarks.

**Strengths:**

This paper is well-motivated, designing a method to improve the ability of a reward model to score responses to prompts while ignoring spurious elements of the response, such as length or some style. They show strong results from including the embeddings from their model during RM training, getting appreciable improvements on standard RM benchmarks.

**Weaknesses:**

I'm curious if you've tested using these RMs during any online RL training, or in other experimental settings like best-of-n rankings, which are generally better correlated with post-online-RL performance? Your method adds a bit of extra required computation per RM inference step if I'm understanding correctly, and I'm curious what the gains vs performance hit looks like.

Also, did you perform any hyperparameter sweeps (mainly for learning rate) for your models? I see you included the settings you used, but these can at times be sensitive to LR especially, so I'd be curious how you chose those settings, and if compute allows, how performance varies.

Also, a general note about the paper, but the number of theorems and mathematical notation at times takes away from the points you're trying to make. I'd recommend simplifying the notation used in the paper, save some space for any other experiments you've run (if relevant), and include the detailed theorems in an appendix.

**Questions:**

Nit: on line 152, I think you meant Llama 3 70B, not 72B?

Can I ask, why did you evaluate on RewardBench 1, when you're using data from RewardBench 2? RewardBench 2 is generally a more difficult benchmark, and includes the ties metric which may be relevant for you (measuring how well reward models handle "tied" responses), so I'm a bit confused why you didn't evaluate on it.

I'm also a bit confused by some of your experimental setup in section 3, specifically in table 1. Can you tell me more about the setup, and why you're measuring how often the model chooses the original response vs a rewritten one? Are the rewritten ones *always* being changed in a negative way (i.e., we always assume the original response is better)? It wasn't clear to me while I was reading the paper.

---

### Note · Authors · 2025-12-04

I have read and agree with the venue's withdrawal policy on behalf of myself and my co-authors.